# Intentional Updates for Streaming Reinforcement Learning

Arsalan Sharifnassab [1]   Mohamed Elsayed [2 3]   Kris De Asis [1]   A. Rupam Mahmood [2 3 4]   Richard S. Sutton [1 2 3 4]

## Abstract

In gradient-based learning, a step size chosen in parameter units does not produce a predictable per-step change in function output. This often leads to instability in the streaming setting (i.e., batch size= 1), where stochasticity is not averaged out and update magnitudes can momentarily become arbitrarily big or small. Instead, we propose *intentional updates*: first specify the *intended outcome* of an update and then solve for the step size that approximately achieves it. This strategy has precedent in online supervised linear regression via the Normalized Least Mean Squares (NLMS) algorithm, which selects a step size to yield a specified change in the function output proportional to the current error. We extend this principle to streaming deep reinforcement learning by defining appropriate intended outcomes: *Intentional TD* aims for a fixed fractional reduction of the TD error, and *Intentional Policy Gradient* aims for a bounded per-step change in the policy, limiting local KL divergence. We propose practical algorithms combining eligibility traces and diagonal scaling. Empirically, these methods yield state-of-the-art streaming performance, frequently performing on par with batch and replay-buffer approaches.[1]

## 1. Introduction

Gradient-based methods learn through a sequence of parameter updates, each computed from the gradient, and scaled by a step size. While this step size dictates how much the weights change, it does not guarantee a consistent change in the function's outputs. Because the relationship between parameters and outputs is complex and gradients vary across data, the updates often become alternately too large or too small—a phenomenon often referred to as overshooting and undershooting (e.g., see Mahmood et al. 2012). This causes instability, especially in streaming learning (i.e., batch size of one), where stochasticity is not averaged out. A prominent example is streaming deep reinforcement learning, where successive per-sample gradients vary sharply in magnitude and direction. This variability makes the fully streaming regime brittle and often causes failure, a phenomenon recently emphasized as a *stream barrier* (Elsayed et al. 2024) in reinforcement learning.

We propose instead to specify the *intended outcome* of an update and then solve for the step size that approximately delivers it. The intended outcome is expressed in the units of what we care about changing, not in the units of the parameters. We instantiate intended outcomes for value learning and policy learning in streaming reinforcement learning, introducing a family of algorithms that employ *intentional updates*. *Intentional TD* aims for a bounded per-step change in the value function by targeting a fractional reduction of the TD error. *Intentional Policy Gradient* aims for a bounded per-step change in the policy by targeting a controlled change in the sampled action's log-probability, a simple streaming proxy for limiting local KL divergence.

This strategy relates closely to normalized update rules in adaptive filtering, such as normalized LMS (Nagumo & Noda 1967; Goodwin & Sin 1984), which scale updates to produce a predictable effect on the output rather than on the weights. In stationary supervised learning, such normalization can alter the effective weighting of samples and the convergent solution. However, our focus is on control problems where targets are inherently non-stationary. In this streaming setting, there is no fixed sampling distribution to converge to; the central requirement is stable tracking. Our results suggest that controlling per-step functional change outweighs stationary fixed-point interpretation, effectively replacing the implicit averaging of replay while retaining fully streaming operation.

To enable intentional updates in streaming deep RL, we develop a practical framework that instantiates multiple algorithms: *Intentional TD*, *Intentional Q-learning*, and

[1]Openmind Research Institute [2]Department of Computing Science, University of Alberta [3]Alberta Machine Intelligence Institute (Amii) [4]CIFAR Canada AI Chair. Correspondence to: Arsalan Sharifnassab <arsalan.sharifnassab@openmindresearch.org>, Mohamed Elsayed <mohamedelsayed@ualberta.ca>.

*Proceedings of the $43^{rd}$ International Conference on Machine Learning*, Seoul, South Korea. PMLR 306, 2026. Copyright 2026 by the author(s).

[1]Code is available at https://github.com/sharifnassab/Intentional_RL

*Intentional Policy Gradient*. These methods yield state-of-the-art streaming performance, often comparable to batch and replay-buffer learning while remaining fully streaming.

**Contributions**

- We propose an intended-outcome principle for step-size selection in streaming gradient-based learning, specifying the intended outcome of an update in the output units, then solving for a step size that approximately achieves it.

- We instantiate this principle for streaming deep reinforcement learning with *Intentional TD*, *Intentional Q-learning*, and *Intentional Policy Gradient*, controlling learning in prediction and policy units rather than parameter units.

- We develop deep RL implementations that integrate eligibility traces and diagonal scaling, and we show robust streaming results across value prediction, continuous control, and discrete-action domains, often comparable to batch and replay-buffer baselines.

## 2. Background

We consider a streaming reinforcement learning setting where an agent learns from a continual sequence of transitions $(s_t, a_t, r_t, s_{t+1})$ without explicit storage of this information (e.g., no replay buffers). A value function is a prediction of future return; for example $V_{\boldsymbol{w}}(s)$ (or $Q_{\boldsymbol{w}}(s, a)$) estimates the conditional expected discounted return from a state (or state–action pair), parameterized by weights $\boldsymbol{w}$. In one-step temporal difference (TD) learning (Sutton 1988; Sutton & Barto 2018), the value function is learnt through minimizing the difference between predictions made at subsequent time steps,

$$\boldsymbol{w}_{t+1} = \boldsymbol{w}_t + \alpha \delta_t \nabla_{\boldsymbol{w}} V_{\boldsymbol{w}_t}(s_t), \tag{1}$$

where $\alpha > 0$ is the step size. The key signal is the TD error, $\delta_t = r_t + \gamma V_{\boldsymbol{w}}(s_{t+1}) - V_{\boldsymbol{w}}(s_t)$, where $\gamma \in [0, 1]$ is the discount factor.

Eligibility traces (Sutton 1988; Sutton & Barto 2018) help facilitate temporal credit assignment in the streaming setting—rather than updating based solely on the current gradient of the value function. We maintain a trace vector $\boldsymbol{z}_t$ representing a decaying memory of past gradients: $\boldsymbol{z}_t = \lambda \gamma \boldsymbol{z}_{t-1} + \nabla_{\boldsymbol{w}} V_{\boldsymbol{w}_t}(s_t)$, where $\lambda \in [0, 1]$ dictates the decay rate. The TD($\lambda$) update is then:

$$\boldsymbol{w}_{t+1} = \boldsymbol{w}_t + \alpha \delta_t \boldsymbol{z}_t. \tag{2}$$

This has an interpretation as interpolating between a one-step TD ($\lambda = 0$) and a Monte Carlo return target ($\lambda = 1$).

For control, policy gradient methods adjust a parameterized policy $\pi_\theta(a|s)$ to increase expected return. The main practical issue is the size of the policy change: small steps in $\theta$ can still produce large changes in action probabilities. Natural-gradient and trust-region methods limit the change in probability space (e.g., Schulman et al. 2015; 2017), but typically add computation and/or rely on batch-style updates.

## 3. Intentional Updates

Intentional updates choose the step size to achieve a specified *intended outcome*, rather than a specified change in the parameters. The intended outcome is defined as a change in a scalar target quantity that represents meaningful progress on the current step. In later sections, this quantity will be a value prediction, an action-value prediction, or a measure of policy change.

Consider any update on network weights $\boldsymbol{w}_t$ that applies a direction $\boldsymbol{d}_t$ computed from the current stream of data,

$$\boldsymbol{w}_{t+1} = \boldsymbol{w}_t + \alpha_t \boldsymbol{d}_t,$$

where $\alpha_t$ is the step size at time $t$. Let $y_t(\boldsymbol{w})$ denote the target quantity we want to control on step $t$ (e.g., $y_t(\boldsymbol{w}) = V_{\boldsymbol{w}}(s_t)$ or $y_t(\boldsymbol{w}) = \log \pi_{\boldsymbol{w}}(a_t|s_t)$). For intentional updates, we specify an intended change $\Delta_t$ in the target quantity,

$$y_t(\boldsymbol{w}_{t+1}) \approx y_t(\boldsymbol{w}_t) + \Delta_t. \tag{3}$$

Using a first-order approximation around $\boldsymbol{w}_t$, we have

$$\begin{aligned} y_t(\boldsymbol{w}_{t+1}) - y_t(\boldsymbol{w}_t) &\approx \nabla y_t(\boldsymbol{w}_t)^\top (\boldsymbol{w}_{t+1} - \boldsymbol{w}_t) \\ &= \alpha_t \nabla y_t(\boldsymbol{w}_t)^\top \boldsymbol{d}_t. \end{aligned} \tag{4}$$

Equating (3) and (4) yields the step size

$$\alpha_t = \frac{\Delta_t}{\nabla y_t(\boldsymbol{w}_t)^\top \boldsymbol{d}_t}. \tag{5}$$

In many learning rules, $\boldsymbol{d}_t$ is proportional to $\nabla y_t(\boldsymbol{w}_t)$, making the step size boil down to $\alpha_t \propto \Delta_t / \|\nabla y_t(\boldsymbol{w}_t)\|_2^2$. In the linear setting, this recovers the classic *Normalized Least Mean Squares* (NLMS) rule (Nagumo & Noda 1967). NLMS makes learning insensitive to feature magnitude; in flat regions (small gradient norm), the step size increases, and in steep regions, it decreases to avoid over-correction.

Intentional updates do not prescribe how $\boldsymbol{d}_t$ is formed; it only uses $\boldsymbol{d}_t$ to compute the step size that achieves the desired change in $y_t$. Finally, (5) is based on a local linear approximation and can produce extreme values when the denominator is small. In practice, we can use standard safeguards, like an $\epsilon$-floor, or a more conservative choice based on the Cauchy-Schwarz inequality ($\boldsymbol{a}^\top \boldsymbol{b} \leqslant \|\boldsymbol{a}\|\|\boldsymbol{b}\|$), which yields $\alpha_t = \Delta_t / \left(\|\boldsymbol{d}_t\|_2 \|\nabla y_t(\boldsymbol{w}_t)\|_2\right)$.

## 4. Intentional TD learning and Q-Learning

We now instantiate an intentional update for value-based learning in prediction and control. The basic pattern is the same in both cases: we specify the intended outcome as a fixed fraction of the current TD error. This makes each update a unit of progress measured in value units.

**TD learning.** Consider the semi-gradient TD(0) update in (1). Intentional TD aims for the following outcome for the prediction

$$V_{\boldsymbol{w}_{t+1}}(s_t) \approx V_{\boldsymbol{w}_t}(s_t) + \eta\,\delta_t, \qquad \eta \in (0,1]. \quad (6)$$

Using (5) with $y_t(\boldsymbol{w}) = V_{\boldsymbol{w}}(s_t)$ and $\boldsymbol{d}_t = \delta_t \nabla_{\boldsymbol{w}} V_{\boldsymbol{w}_t}(s_t)$ yields

$$\alpha_t = \frac{\eta}{\|\nabla_{\boldsymbol{w}} V_{\boldsymbol{w}_t}(s_t)\|_2^2}. \quad (7)$$

Under the same first-order approximation used to derive (7), this choice contracts the momentary error

$$V_{\boldsymbol{w}_{t+1}}(s_t) - V_t^{\text{target}} \approx (1 - \eta)\Big(V_{\boldsymbol{w}_t}(s_t) - V_t^{\text{target}}\Big),$$

where $V_t^{\text{target}} \doteq r_t + \gamma V_{\boldsymbol{w}_t}(s_{t+1})$.

**Q-learning control.** For control we apply this construction to action values by choosing $y_t(\boldsymbol{w}) = Q_{\boldsymbol{w}}(s_t, a_t)$ and

$$\delta_t \doteq r_t + \gamma \max_{a'} Q_{\boldsymbol{w}_t}(s_{t+1}, a') - Q_{\boldsymbol{w}_t}(s_t, a_t).$$

Intentional Q-learning again targets a fixed fractional reduction of the current error at the sampled pair $(s_t, a_t)$

$$Q_{\boldsymbol{w}_{t+1}}(s_t, a_t) - Q_{\boldsymbol{w}_t}(s_t, a_t) \approx \eta\,\delta_t,$$

which yields

$$\alpha_t = \frac{\eta}{\|\nabla_{\boldsymbol{w}} Q_{\boldsymbol{w}_t}(s_t, a_t)\|_2^2}.$$

**Handling different scales with RMSProp.** Since different gradient entries can have vastly different scales, we adopt an RMSProp-style entry-wise normalization (Tieleman & Hinton 2012), where we maintain an exponential moving average, $\boldsymbol{\nu}_t$, of squared gradients, and apply entry-wise scaling $\boldsymbol{\rho}_t = 1/\sqrt{\boldsymbol{\nu}_t + \epsilon}$. With this scaling, the TD(0) update in (1), turns into $\boldsymbol{w}_{t+1} = \boldsymbol{w}_t + \alpha_t \delta_t \boldsymbol{\rho}_t \nabla_{\boldsymbol{w}} V_{\boldsymbol{w}_t}(s_t)$. Applying (5) to this update, while using the same intended outcome as in (6), we obtain an alternative formula to (7) for the step-size: $\alpha_t = \eta/\langle \nabla_{\boldsymbol{w}} V_{\boldsymbol{w}_t}(s_t),\, \boldsymbol{\rho}_t \nabla_{\boldsymbol{w}} V_{\boldsymbol{w}_t}(s_t)\rangle$.

**Intentional updates with eligibility traces.** We consider TD($\lambda$) update (2) with eligibility traces, and integrate it with an RMSprop-style entry-wise scaling $\boldsymbol{\rho}_t$ discussed in the previous paragraph,

$$\boldsymbol{w}_{t+1} = \boldsymbol{w}_t + \alpha_t \delta_t \boldsymbol{\rho}_t \boldsymbol{z}_t. \quad (8)$$

where $\boldsymbol{g}_t = \nabla_{\boldsymbol{w}} V_{\boldsymbol{w}_t}(s_t)$ and $\lambda \in [0,1]$ is the trace parameter.

With eligibility traces, a single update induces coordinated changes across a fading history of predictions; therefore, the intended outcome must be defined in terms of an *aggregate* change over past predictions rather than the change in the current prediction only. We choose $\alpha_t$ so that the discounted root-mean-square-change in recent predictions is proportional to $|\delta_t|$:

$$\sqrt{\sum_{\tau \leqslant t} (\lambda\gamma)^{t-\tau}\Big(V_{\boldsymbol{w}_{t+1}}(s_\tau) - V_{\boldsymbol{w}_t}(s_\tau)\Big)^2} \;\approx\; \eta\,|\delta_t|. \quad (9)$$

Using a first-order approximation, the change in an earlier prediction $V_{\boldsymbol{w}}(s_\tau)$ caused by the update (8) is

$$V_{\boldsymbol{w}_{t+1}}(s_\tau) - V_{\boldsymbol{w}_t}(s_\tau) \approx \nabla_{\boldsymbol{w}} V_{\boldsymbol{w}_t}(s_\tau)^\top (\boldsymbol{w}_{t+1} - \boldsymbol{w}_t)$$
$$= \alpha_t \delta_t \langle \boldsymbol{g}_\tau, \boldsymbol{\rho}_t \boldsymbol{z}_t\rangle,$$

where $\boldsymbol{g}_\tau \doteq \nabla_{\boldsymbol{w}} V_{\boldsymbol{w}_\tau}(s_\tau)$. Substituting this into the aggregate objective (9) gives

$$\sum_{\tau \leqslant t} (\lambda\gamma)^{t-\tau}\Big(V_{\boldsymbol{w}_{t+1}}(s_\tau) - V_{\boldsymbol{w}_t}(s_\tau)\Big)^2$$
$$\approx \alpha_t^2 \delta_t^2 \sum_{\tau \leqslant t} (\lambda\gamma)^{t-\tau}\langle \boldsymbol{g}_\tau, \boldsymbol{\rho}_t \boldsymbol{z}_t\rangle^2$$
$$\leqslant \alpha_t^2 \delta_t^2 \langle \boldsymbol{z}_t, \boldsymbol{\rho}_t \boldsymbol{z}_t\rangle \sum_{\tau \leqslant t} (\lambda\gamma)^{t-\tau}\langle \boldsymbol{g}_\tau, \boldsymbol{\rho}_t \boldsymbol{g}_\tau\rangle,$$
$$(10)$$

where the last inequality is a conservative approximation that follows from the Cauchy–Schwarz inequality.

Let

$$\sigma_\tau \doteq \langle \boldsymbol{g}_\tau, \boldsymbol{\rho}_\tau \boldsymbol{g}_\tau\rangle, \qquad \bar{\sigma}_t \doteq \sum_{\tau \leqslant t} (\lambda\gamma)^{t-\tau}\sigma_\tau, \quad (11)$$

where in practice $\bar{\sigma}_t$ is maintained by a discounted running estimate (Section 6). Choosing

$$\alpha_t = \frac{\eta}{\sqrt{\bar{\sigma}_t \langle \boldsymbol{\rho}_t \boldsymbol{z}_t, \boldsymbol{z}_t\rangle}} \quad (12)$$

makes the right-hand side of (10) equal to $\eta^2 \delta_t^2$, and therefore satisfies the aggregate targeting condition (9) in a conservative sense (with $\lesssim$ in place of $\approx$). When $\lambda = 0$ and $\boldsymbol{\rho}_t = \mathbf{1}$, $\boldsymbol{z}_t = \boldsymbol{g}_t$ and (12) reduces to (7). The same construction applies to Q-learning by replacing $V_{\boldsymbol{w}}(s)$ and $\boldsymbol{g}_t$ with $Q_{\boldsymbol{w}}(s, a)$ and $\nabla_{\boldsymbol{w}} Q_{\boldsymbol{w}}(s, a)$. Refer to Appendix A for detailed derivation.

To build intuition, consider a tempting and naive—but ultimately incorrect— extension of (7) to TD($\lambda$) in (8) by setting $\alpha_t = \eta/\langle \boldsymbol{\rho}_t \boldsymbol{z}_t, \boldsymbol{z}_t\rangle$. This, however, is not the

right way and entails some important problems. In particular, it does not scale correctly with the trace, as it makes the effective update shrink as the trace grows. If recent gradients align so that $\boldsymbol{g}_\tau \simeq \boldsymbol{g}_t$ for $\tau \leqslant t$, then the trace grows $\boldsymbol{z}_t \simeq \boldsymbol{g}_t/(1-\lambda)$ and the update becomes $\alpha_t\delta_t\boldsymbol{\rho}_t\boldsymbol{z}_t \simeq \eta(1-\lambda)\delta_t\boldsymbol{\rho}_t\boldsymbol{g}_t/\langle\boldsymbol{\rho}_t\boldsymbol{g}_t, \boldsymbol{g}_t\rangle$. If instead past gradients are negligible so that $\boldsymbol{z}_t \simeq \boldsymbol{g}_t$, the same rule yields $\alpha_t\delta_t\boldsymbol{\rho}_t\boldsymbol{z}_t \simeq \eta, \delta_t, \boldsymbol{\rho}_t\boldsymbol{g}_t/\langle\boldsymbol{\rho}_t\boldsymbol{g}_t, \boldsymbol{g}_t\rangle$, larger by a factor $1/(1-\lambda)$. Thus, the normalization $\alpha_t = \eta/\langle\boldsymbol{\rho}_t\boldsymbol{z}_t, \boldsymbol{z}_t\rangle$ reverses the scaling we want: longer, more coherent traces lead to smaller per-state updates. In contrast, (12) normalizes by an estimate of the aggregate induced change, so these two cases produce comparable updates and, in the linear case, yield the intended propagation of the TD error over past states, with scale $\eta(1-\lambda)$ and decayed by $\lambda$.

## 5. Intentional Policy Learning

For value learning, Intentional TD targets a controlled change in a prediction toward a bootstrap target. For policy learning, there is no scalar target to close a gap with. Instead, we target a controlled per-step change in the policy itself. Each update should produce a comparable amount of behavioral change over time proportional to its advantage, despite changing gradient scales and representations.

**A unit of policy change.** A convenient local measure of policy change at state $s_t$ is the change in log-probability at the sampled action, $\Delta\log\pi_t \doteq \log\pi_{\boldsymbol{\theta}_{t+1}}(a_t|s_t) - \log\pi_{\boldsymbol{\theta}_t}(a_t|s_t)$. This quantity is easy to compute online and is directly connected to probability change: for small steps, a constant $\Delta\log\pi_t$ corresponds to a roughly constant multiplicative change in $\pi(a_t|s_t)$. It is also related to the state-conditional KL divergence used in trust-region and PPO-style methods as a measure of policy shift (e.g., see Schulman et al. 2015; 2017). In particular, for small changes,

$$D_{\mathrm{KL}}\big(\pi_{\boldsymbol{\theta}_t}(\cdot|s_t)\,\|\,\pi_{\boldsymbol{\theta}_{t+1}}(\cdot|s_t)\big)$$
$$\approx \frac{1}{2}\,\mathbb{E}_{a\sim\pi_{\boldsymbol{\theta}_t}(\cdot|s_t)}\Big[\big(\Delta\log\pi(a|s_t)\big)^2\Big], \quad (13)$$

so controlling typical magnitudes of $\Delta\log\pi$ provides a simple proxy for controlling local KL change. (see Appendix A for details). Our method targets the sampled $\Delta\log\pi_t$ directly; this is not an exact KL constraint, but it provides an inexpensive, per-step unit of policy change.

**Intentional Policy Gradient step size.** We start from the likelihood-ratio policy-gradient update

$$\boldsymbol{\theta}_{t+1} = \boldsymbol{\theta}_t + \alpha_t A_t \boldsymbol{g}_t, \qquad \boldsymbol{g}_t \doteq \nabla_{\boldsymbol{\theta}}\log\pi_{\boldsymbol{\theta}_t}(a_t|s_t),$$

where $A_t$ is an advantage estimate (e.g., see Sutton & Barto 2018; Sutton et al. 1999). We choose the intended outcome as a per-step change in $\log\pi_{\boldsymbol{\theta}}(a_t|s_t)$ proportional to

advantage with typical magnitude $\eta$:

$$\log\pi_{\boldsymbol{\theta}_{t+1}}(a_t|s_t) - \log\pi_{\boldsymbol{\theta}_t}(a_t|s_t) \approx \eta\frac{A_t}{\bar{A}_t}, \quad (14)$$

where $\bar{A}_t$ is a running scale for $|A_t|$ (defined below). This keeps the long-run average magnitude of policy change near $\eta$, while the per-step change remains proportional to $A_t$. The normalization, therefore, sets the typical update scale rather than imposing a hard cap, so rare transitions with unusually large advantage can still induce correspondingly large policy changes.

Using a first-order approximation,

$$\Delta\log\pi_t \approx \nabla_{\boldsymbol{\theta}}\log\pi_{\boldsymbol{\theta}_t}(a_t|s_t)^\top(\boldsymbol{\theta}_{t+1}-\boldsymbol{\theta}_t) = \alpha_t A_t\|\boldsymbol{g}_t\|_2^2, \tag{15}$$

and equating (15) with (14) gives the step size

$$\alpha_t = \frac{\eta}{\bar{A}_t\|\boldsymbol{g}_t\|_2^2}. \tag{16}$$

With (16), the sampled log-probability change satisfies $\Delta\log\pi_t \approx \eta A_t/\bar{A}_t$, so $\eta$ directly sets the typical per-step magnitude of policy change, and by (13) it also sets the scale of local KL change in the small-step regime (detailed discussion in Appendix A).

We estimate $\bar{A}_t$ by an exponential average of $|A_t|$,

$$\bar{A}_t = \bar{A}_{t-1} + \frac{1-\beta_{\mathrm{norm}}}{1-\beta_{\mathrm{norm}}^t}\Big(|A_t| - \bar{A}_{t-1}\Big).$$

**Traces and diagonal normalization.** As in Section 4, we use eligibility traces and RMSProp-style diagonal normalization in our streaming implementations (Section 6). These mechanisms modify the update direction (e.g., replacing $\boldsymbol{g}_t$ by a trace $\boldsymbol{z}_t$ and applying an entry-wise preconditioner $\boldsymbol{\rho}_t$), and the intentional update then chooses $\alpha_t$ using the same principle applied to the resulting direction.

**A note on sample-dependent step sizes.** Our instances of intentional updates choose $\alpha_t$ from the current sample, which can change the update *in expectation*. For value learning, this mainly reweights states, changing how quickly different parts of the stream are tracked. For policy learning, because the sample includes the action, action-dependent normalization can also reweight actions and thereby change the expected update direction at each state. In our experiments, this did not hinder learning, but it is useful to distinguish the generally benign state reweighting in prediction from the potentially consequential action reweighting in control. Appendix B discusses this phenomenon.

## 6. Algorithms

This section summarizes the practical streaming optimizers used in our experiments. We present three algorithms: Intentional TD($\lambda$) for value prediction, Intentional Q($\lambda$) for

---

**Algorithm 1** Intentional TD($\lambda$)

---

**Require:** $\lambda \in [0,1), \gamma \in [0,1], \eta > 0, \beta_\nu \in [0,1)$
**Default values:** $\beta_\nu = 0.999, \epsilon = 10^{-8}$
**Initialize:** $\boldsymbol{w}_1, \boldsymbol{z}_0 = \boldsymbol{0}, \bar\sigma_0 = 0, \boldsymbol{\nu}_0 = \boldsymbol{0}$
**for** $t = 1, 2, \ldots$
   Observe transition $(s_t, r_t, s_{t+1})$
   $\delta_t \leftarrow r_t + \gamma V_{\boldsymbol{w}_t}(s_{t+1}) - V_{\boldsymbol{w}_t}(s_t)$
   $\tilde\delta_t \leftarrow \text{Clip}(\delta_t)$                $\triangleright$ compute using (17)
   $\boldsymbol{g}_t \leftarrow \nabla_{\boldsymbol{w}} V_{\boldsymbol{w}_t}(s_t)$
   $\boldsymbol{\nu}_t \leftarrow \boldsymbol{\nu}_{t-1} + \frac{1-\beta_\nu}{1-\beta_\nu^t}(\boldsymbol{g}_t^2 - \boldsymbol{\nu}_{t-1})$
   $\boldsymbol{\rho}_t \leftarrow (\sqrt{\boldsymbol{\nu}_t} + \epsilon)^{-1}$
   $\sigma_t \leftarrow \langle \boldsymbol{\rho}_t \boldsymbol{g}_t, \boldsymbol{g}_t \rangle$
   $\bar\sigma_t \leftarrow \bar\sigma_{t-1} + \frac{1-\lambda\gamma}{1-(\lambda\gamma)^t}(\sigma_t - \bar\sigma_{t-1})$
   $\boldsymbol{z}_t \leftarrow \lambda\gamma \boldsymbol{z}_{t-1} + \boldsymbol{g}_t$          $\triangleright$ eligibility trace
   $\alpha_t \leftarrow \eta / \sqrt{\bar\sigma_t \langle \boldsymbol{\rho}_t \boldsymbol{z}_t, \boldsymbol{z}_t \rangle}$
   $\boldsymbol{w}_{t+1} \leftarrow \boldsymbol{w}_t + \alpha_t \tilde\delta_t \boldsymbol{\rho}_t \boldsymbol{z}_t$

---

**Algorithm 2** Intentional Q($\lambda$)

---

**Require:** $\lambda \in [0,1), \gamma \in [0,1], \eta > 0, \beta_\nu \in [0,1)$
**Default values:** $\beta_\nu = 0.999, \epsilon = 10^{-8}$
**Initialize:** $\boldsymbol{w}_1, \boldsymbol{z}_0 = \boldsymbol{0}, \bar\sigma_0 = 0, \boldsymbol{\nu}_0 = \boldsymbol{0}$
**for** $t = 1, 2, \ldots$
   Observe transition $(s_t, a_t, r_t, s_{t+1}, a_{t+1})$
   $\delta_t \leftarrow r_t + \gamma \max_{a'} Q_{\boldsymbol{w}_t}(s_{t+1}, a') - Q_{\boldsymbol{w}_t}(s_t, a_t)$
   $\tilde\delta_t \leftarrow \text{Clip}(\delta_t)$              $\triangleright$ compute using (17)
   $\boldsymbol{g}_t \leftarrow \nabla_{\boldsymbol{w}} Q_{\boldsymbol{w}_t}(s_t, a_t)$
   $\boldsymbol{\nu}_t \leftarrow \boldsymbol{\nu}_{t-1} + \frac{1-\beta_\nu}{1-\beta_\nu^t}(\boldsymbol{g}_t^2 - \boldsymbol{\nu}_{t-1})$
   $\boldsymbol{\rho}_t \leftarrow (\sqrt{\boldsymbol{\nu}_t} + \epsilon)^{-1}$
   $\sigma_t \leftarrow \langle \boldsymbol{\rho}_t \boldsymbol{g}_t, \boldsymbol{g}_t \rangle$
   $\bar\sigma_t \leftarrow \bar\sigma_{t-1} + \frac{1-\lambda\gamma}{1-(\lambda\gamma)^t}(\sigma_t - \bar\sigma_{t-1})$
   $\boldsymbol{z}_t \leftarrow \lambda\gamma \boldsymbol{z}_{t-1} + \boldsymbol{g}_t$          $\triangleright$ eligibility trace
   $\alpha_t \leftarrow \eta / \sqrt{\bar\sigma_t \langle \boldsymbol{\rho}_t \boldsymbol{z}_t, \boldsymbol{z}_t \rangle}$
   $\boldsymbol{w}_{t+1} \leftarrow \boldsymbol{w}_t + \alpha_t \tilde\delta_t \boldsymbol{\rho}_t \boldsymbol{z}_t$

---

**Algorithm 3** Intentional Policy Gradient

---

**Require:** $\lambda \in [0,1), \gamma \in [0,1], \eta > 0, \beta_{\text{norm}} \in [0,1),$
   $\beta_\nu \in [0,1)$, entropy coefficient $\xi \geq 0$,
**Default values:** $\beta_{\text{norm}} = 0.9998, \beta_\nu = 0.999, \epsilon = 10^{-8}$
**Initialize:** $\boldsymbol{\theta}_1, \bar\sigma_0 = \bar{A}_0 = 0, \boldsymbol{z}_0 = \boldsymbol{\nu}_0 = \boldsymbol{0}$
**for** $t = 1, 2, \ldots$
   Observe $(s_t, a_t, r_t, s_{t+1})$
   Get TD error $\delta_t$
   $A_t \leftarrow \text{Clip}(\delta_t)$             $\triangleright$ compute using (17)
   $\bar{A}_t \leftarrow \bar{A}_{t-1} + \frac{1-\beta_{\text{norm}}}{1-\beta_{\text{norm}}^t}(|A_t| - \bar{A}_{t-1})$
   $\tilde{A}_t \leftarrow A_t / \bar{A}_t$
   $\boldsymbol{g}_t \leftarrow \nabla_{\boldsymbol{\theta}}\Big(\log \pi_{\boldsymbol{\theta}_t}(a_t|s_t) + \xi\, \text{entropy}\,\big(\pi(\cdot|s_t)\big) \text{sign}(\tilde{A}_t)\Big)$
   $\boldsymbol{\nu}_t \leftarrow \boldsymbol{\nu}_{t-1} + \frac{1-\beta_\nu}{1-\beta_\nu^t}(\boldsymbol{g}_t^2 - \boldsymbol{\nu}_{t-1})$
   $\boldsymbol{\rho}_t \leftarrow (\sqrt{\boldsymbol{\nu}_t} + \epsilon)^{-1}$
   $\sigma_t \leftarrow \langle \boldsymbol{\rho}_t \boldsymbol{g}_t, \boldsymbol{g}_t \rangle$
   $\bar\sigma_t \leftarrow \bar\sigma_{t-1} + \frac{1-\lambda\gamma}{1-(\lambda\gamma)^t}(\sigma_t - \bar\sigma_{t-1})$
   $\boldsymbol{z}_t \leftarrow \lambda\gamma \boldsymbol{z}_{t-1} + \boldsymbol{g}_t$
   $\alpha_t \leftarrow \eta / \sqrt{\bar\sigma_t \langle \boldsymbol{\rho}_t \boldsymbol{z}_t, \boldsymbol{z}_t \rangle}$
   $\boldsymbol{\theta}_{t+1} \leftarrow \boldsymbol{\theta}_t + \alpha_t \tilde{A}_t \boldsymbol{\rho}_t \boldsymbol{z}_t$

---

control, and Intentional Policy Gradient for policy optimization. All three share a similar structure: compute a learning signal $\delta_t$, form an update direction (a gradient or an eligibility trace), apply a diagonal normalization, and choose a step size by the corresponding intentional update rule. The complete algorithms are presented in Algorithms 1, 2, and 3.

To reduce sensitivity to parameter scales, we apply an RMSProp-style diagonal scaling $\boldsymbol{\rho}_t$, defined as the inverse square root of an exponentially decayed average of past squared gradients ($\boldsymbol{\nu}_t$). This incorporates an Adam-style bias correction $(1 - \beta_\nu^t)$ for small $t$.

In all three algorithms, we use eligibility traces, $\boldsymbol{z}_t = \lambda\gamma \boldsymbol{z}_{t-1} + \boldsymbol{g}_t$, and update in the preconditioned trace direction $\boldsymbol{\rho}_t \boldsymbol{z}_t$. The step size is chosen to satisfy the aggregate targeting objective introduced in Section 4. The resulting conservative step size is given in (12).

We also incorporate $\delta_t$ clipping to improve stability under rare but large TD-error outliers. Instead of clipping $\delta$ to the range $[-1, 1]$, proposed in Elsayed et al. (2024)[2], we clip $\delta$ to a constant multiple of its long-term root mean square:

$$\hat\delta_t = \hat\delta_{t-1} + \frac{1 - \beta_{\text{clip}}}{1 - \beta_{\text{clip}}^t}(\tilde\delta_t^2 - \hat\delta_{t-1})$$

$$\tilde\delta_t = \text{sign}(\delta_t)\,\min\left(|\delta_t|, C\sqrt{\hat\delta_t}\right), \tag{17}$$

where $C$ (set to 20) and $\beta_{\text{clip}}$ (set to 0.9998) are meta-parameters. This makes the clipping invariant to the scale of $\delta_t$. That is, if we multiply $\delta_\tau$ by a constant for all times $\tau$, then $\tilde\delta_t$ would scale proportionally, as opposed to the constant range clipping.

In Algorithm 3, following Elsayed et al. (2024), the entropy gradient is included in $\boldsymbol{g}_t$, which is then accumulated in the eligibility trace $\boldsymbol{z}$, and multiplied by $\delta$. This avoids the need for a separate backward pass to compute the entropy gradient. Alternatively, the entropy gradient could bypass the trace and $\delta$ to be added directly to the current update. Empirically, we observed a negligible difference between these two approaches in MuJoCo.

## 7. Experiments

We evaluate the proposed algorithms in the streaming deep RL setting. Across continuous control (MuJoCo and DM Control) and discrete-action control (MinAtar and Atari), the intentional update algorithms yield stable learning

---

[2]The update rule in Elsayed et. (2024) is $\propto \text{clip}_{[-1,1]}(\delta)\frac{\boldsymbol{z}}{\|\boldsymbol{z}\|_1}$.

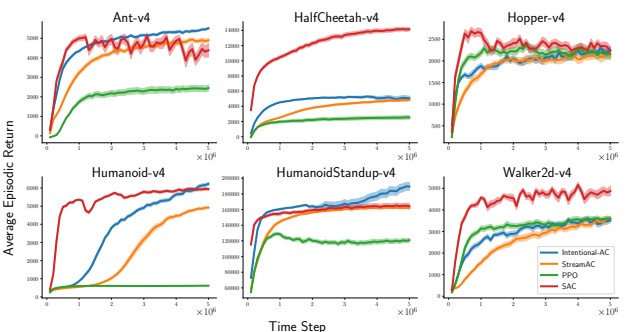

*Figure 1.* Average episodic return versus environment steps on MuJoCo environments.

and strong final performance. Moreover, a single meta-parameter setting per algorithm transfers across environments within each benchmark family, and the resulting agents are less dependent on auxiliary stabilization than prior streaming methods.

### 7.1. Setup

We use the benchmark suites, agent codebase, and evaluation protocol of Elsayed et al. (2024). All experiments are fully streaming: each environment step produces at most one update, with no replay and no mini-batches. Unless stated otherwise, we use the same network architectures and preprocessing as Elsayed et al. (2024). For time-limited episodic tasks, we consider a continual approximation by bootstrapping at truncation (timeouts) instead of bootstrapping at true terminals.

Each curve reports the mean over 30 independent runs, with shaded regions showing 95% confidence intervals. All details, including meta-parameters, environment lists, and implementation details, are provided in Appendix D.

### 7.2. Continuous control: MuJoCo

We train a streaming actor–critic on MuJoCo Gym tasks. We use Intentional TD($\lambda$) described in Algorithm 1 for the critic, and we use Intentional Policy Gradient described in Algorithm 3 for the actor. We refer to the resulting actor–critic combination as *Intentional AC*. Figure 1 shows that Intentional AC consistently stabilizes learning and improves performance relative to the corresponding streaming baseline, while using one shared setting across tasks. In several environments, Intentional AC closes most of the gap between streaming learning and replay-based training.

### 7.3. Continuous control: DM Control Suite

We repeat the actor–critic evaluation on the DM Control tasks. Figure 2 shows the same pattern as MuJoCo: Intentional AC is stable across tasks and seeds, and outperforms the streaming baselines, again with one shared parameter setting across the suite. Performance still differs across

tasks under the shared setting. For example, Finger-turn was unstable across many seeds, and with smaller scales, $\eta_{\text{actor}} = 0.01$ and $\eta_{\text{critic}} = 0.1$, it improved substantially. This suggests good transfer of the shared scales across the suite, while some tasks still benefit from further tuning.

### 7.4. Discrete-action control: Atari and MinAtar

We evaluate Intentional Q-learning control on Atari and MinAtar under the streaming setup of Elsayed et al. (2024). Figures 3 and 4 show that Intentional Q-learning yields reliable learning across games and achieves a performance competitive with batch methods.

### 7.5. Value prediction under a fixed policy

To isolate prediction from control, we evaluate streaming value learning under a fixed behavior policy. For each MuJoCo task, we first train an actor with the StreamAC algorithm of Elsayed et al. (2024), for 5 million environment steps. We then freeze the policy and train a critic from scratch on the resulting stream.

Performance is measured by the root mean squared prediction error,

$$\sqrt{\mathbb{E}_s\left[\left(V_{\boldsymbol{w}_t}(s) - G(s)\right)^2\right]}, \quad (18)$$

where $G(s)$ is the Monte Carlo discounted return from $s$ under the fixed policy (estimated from rollouts) and the expectation is over states visited by that policy. Figure 5 compares Intentional TD($\lambda$) to StreamTD($\lambda$) (Elsayed et al. 2024). Intentional TD($\lambda$) yields lower error throughout learning and is notably less sensitive to gradient scale.

### 7.6. Ablations

We run two ablation suites. First, we remove individual components from the full intentional update to assess their significance (Figure 6). Specifically, we tested the impacts of: removing diagonal scaling (no RMSProp), removing $\delta$-clipping (no clipping), replacing adaptive $\delta$-clipping (17) with a simple clipping to the range $[-1, 1]$ (non-adaptive clipping), and replacing the stepsize $\alpha_t = \eta/\sqrt{\bar{\sigma}_t \langle \boldsymbol{\rho}_t \boldsymbol{z}_t, \boldsymbol{z}_t \rangle}$ with the naive choice $\alpha_t = \eta/\langle \boldsymbol{\rho}_t \boldsymbol{z}_t, \boldsymbol{z}_t \rangle$ discussed in the last paragraph of Section 4 (no sigma). The main finding is that performance remains competitive when most of these components are removed, showing that the primary driver of performance is the intentional scaling rather than auxiliary components. While the diagonal normalization and the $\sigma$ term improve results, the $\delta$-clipping appears to have a negligible impact in this ablation. We nonetheless recognize the role of $\delta$-clipping in preventing propagation of large value errors, and believe it enhances robustness across broader benchmarks and settings.

Second, we follow the StreamX components ablation in El-

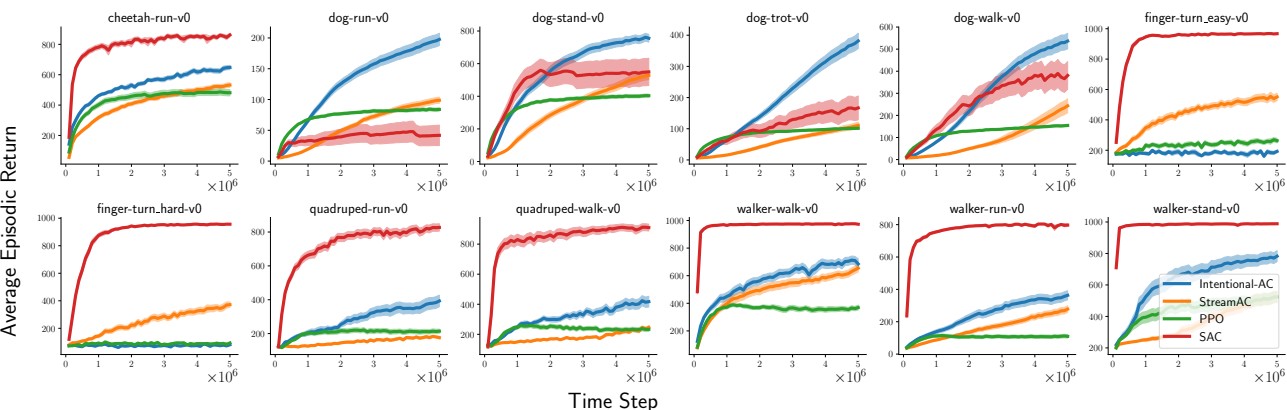

*Figure 2.* DM Control Suite streaming actor–critic. Average episodic return versus environment steps.

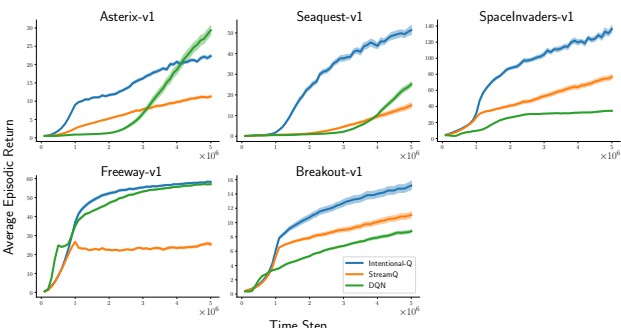

*Figure 3.* Average score versus environment frames on MinAtar environments.

sayed et al. (2024), and remove reward scaling, observation normalization, and sparse initialization while keeping the intentional update fixed (Table 1). Intentional AC shows more robustness to the removal of these preconditioners, compared to the StreamAC baseline. More details and complementary results are presented in Appendix F.

### 7.7. Inner Workings: Fidelity of Linear Approximation and Numerical Robustness

In this experiment, we tested whether the actual changes produced by the update are close to the intended changes. The potential mismatch between intended change and actual change is due to linear approximation (e.g., (4)) involved in the derivations of Intentional step-sizes. For value updates, we measure the change in $V(s_t)$; for policy updates, we measure the change in $\log \pi(a_t \mid s_t)$. We report statistics of the ratio

$$\frac{\text{actual change}}{\text{intended change}}$$

over 5M training steps, and averaged over 30 runs. In this experiment, we disable eligibility traces and set $\lambda = 0$, because the step-size rule with traces involves extra conservative approximations that would make the fidelity measurement harder to interpret. Table 2 shows that the ratio remains close to 1, indicating that the realized change closely

matches the intended change in the regime studied here.

In another experiment, we tested whether intentional scaling substantially increases the variability of realized parameter updates. This matters because flat regions can produce very small denominators, and hence unusually large step sizes. We define the effective update size as

$$\text{effective update} = \frac{\|w_{t+1} - w_t\|_2}{|\delta_t|},$$

where dividing by $|\delta_t|$ helps factor out variation in target magnitude. We then summarize variability by

$$\text{effective update ratio} = \frac{\text{99th percentile of effective updates}}{\text{average of effective updates}},$$

where both statistics are computed over 5M training steps. We compare two versions of the same algorithm: (i) the intentional update, and (ii) the same method with $\alpha = 1$, that is, without intentional scaling but with all other components unchanged.

Table 3 provides effective update ratios, averaged over three MuJoCo environments (Ant, Humanoid, and Humanoid-Standup), and over 30 independent runs per environment. The results suggest that intentional scaling does not by itself create extreme realized updates in the regime studied here.

## 8. Related Work

Our work builds on recent advances in fully streaming deep reinforcement learning, where the agent updates from each new transition immediately, without replay buffers, target networks, or mini-batches. Elsayed et al. (2024) showed that deep RL in this regime often fails (the *stream barrier*) and introduced the StreamX family of algorithms (e.g., StreamQ($\lambda$) and StreamAC($\lambda$) agents) that learn reliably with a shared set of hyperparameters. Closely related work has also shown that deep policy-gradient methods can be made to work without replay or batch updates by combining incremental updates with careful normalization and scaling (e.g., see Vasan et al. 2024). More broadly, eligibility

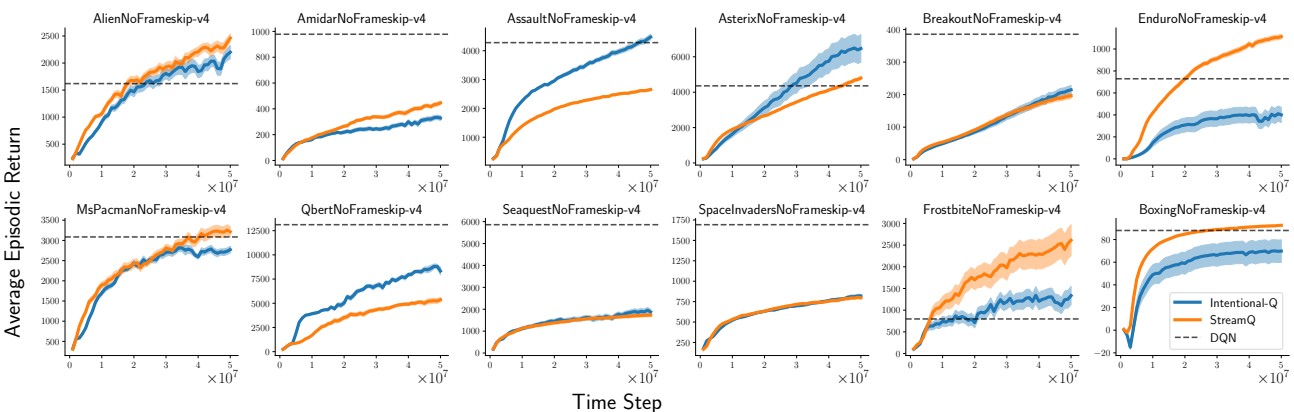

*Figure 4.* Atari streaming control. Average score versus environment frames.

| Environment | Standard | | no SparseInit | | no ScaledReward | | no InputNormalization | | no LayerNorm | |
|---|---|---|---|---|---|---|---|---|---|---|
| | Int. AC | StreamAC | Int. AC | StreamAC | Int. AC | StreamAC | Int. AC | StreamAC | Int. AC | StreamAC |
| Ant | **5513**$_{+54}$ | 4898$_{+84}$ | **3818**$_{+133}$ | 2039$_{+193}$ | **5423**$_{+62}$ | 558$_{+37}$ | **4509**$_{+68}$ | 3604$_{+66}$ | 2323$_{+78}$ | **2541**$_{+79}$ |
| HalfCheetah | **5064**$_{+288}$ | 4830$_{+128}$ | **4513**$_{+725}$ | 2959$_{+207}$ | **3986**$_{+433}$ | 573$_{+117}$ | **4274**$_{+98}$ | 2503$_{+127}$ | **2691**$_{+158}$ | 2666$_{+77}$ |
| Hopper | **2254**$_{+71}$ | 2113$_{+91}$ | **2001**$_{+107}$ | 1695$_{+143}$ | **2094**$_{+91}$ | 295$_{+30}$ | 1067$_{+74}$ | **1092**$_{+78}$ | **1696**$_{+167}$ | 737$_{+131}$ |
| Humanoid | **6227**$_{+80}$ | 4920$_{+51}$ | **2503**$_{+246}$ | 610$_{+10}$ | **5534**$_{+119}$ | 1011$_{+54}$ | **718**$_{+25}$ | 382$_{+1}$ | **2776**$_{+311}$ | 651$_{+11}$ |
| HumanoidStandup | **189k**$_{+5.1k}$ | 162k$_{+1.2k}$ | **221k**$_{+5.7k}$ | 135k$_{+3.2k}$ | **235k**$_{+5.9k}$ | 128k$_{+4.9k}$ | **149k**$_{+2.0k}$ | 91k$_{+4.0k}$ | **164k**$_{+0.8k}$ | 149k$_{+1.4k}$ |
| Walker2d | 3515$_{+113}$ | **3590**$_{+110}$ | **3148**$_{+178}$ | 1469$_{+135}$ | **3022**$_{+140}$ | 180$_{+25}$ | **2152**$_{+142}$ | 1193$_{+73}$ | **1631**$_{+166}$ | 172$_{+160}$ |

*Table 1.* Ablations of robustness to StreamX stabilizers in MuJoCo environments (v4). We report mean return $\pm$ standard error.

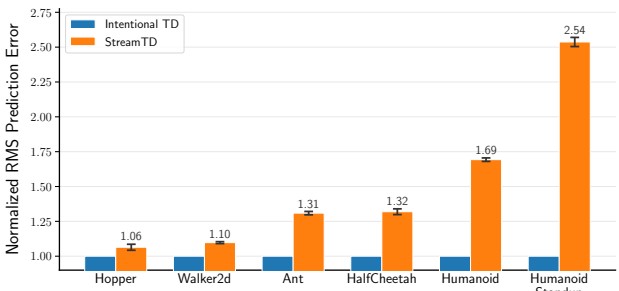

*Figure 5.* Averaged value prediction error under a fixed policy normalized by the prediction error of Intentional TD.

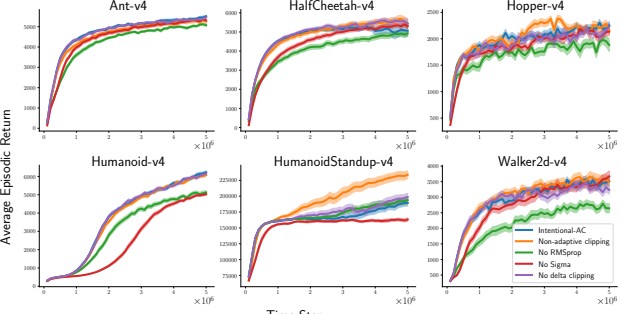

*Figure 6.* Ablations of Intentional AC components.

traces and true-online variants have long been motivated as a way to retain the streaming characteristic of TD methods while improving temporal credit assignment (e.g., see Sutton 1988; van Seijen et al. 2016; van Hasselt et al. 2021).

Targeting an intended outcome is also connected to classical normalized step-size ideas. In supervised learning, the normalized LMS family scales the step size by local sensitivity so that each update produces a more predictable change in the *output* rather than in the weights (e.g., see Widrow & Hoff 1960; Nagumo & Noda 1967; Haykin 2002). Polyak step sizes are another example of choosing step sizes in units of objective reduction, setting $\alpha_t$ proportional to a gap-to-optimum divided by a squared gradient norm (Polyak 1987; Loizou et al. 2021); they require access to an optimal value (or a proxy), which is typically unavailable in reinforcement learning, but they share the same general theme of selecting

step sizes in *function* units rather than parameter units.

A third line of work derives explicit step-size bounds for temporal-difference learning. Dabney & Barto (2012) give adaptive bounds for online TD with linear function approximation, motivated by preventing divergence and overly aggressive updates. More recently, SwiftTD revisits overshooting control and step-size limiting in TD-style updates, again with an emphasis on stability guarantees and practical robustness (Javed et al. 2024). StreamX's Overshooting-bounded Gradient Descent (ObGD) fits this family: it chooses the step size by detecting when a tentative update would "overshoot" according to a sign-change test on the TD error after the update (Elsayed et al. 2024).

Our instances of intentional updates differ from ObGD in what they control. Intentional TD targets the change in the

*Table 2.* Update fidelity ratio (actual change / intended change).

*(a)* Policy update fidelity ratio.

| Environment | Std | 1st Percentile | 99st Percentile |
|---|---|---|---|
| Ant | 0.016 | 0.960 | 1.029 |
| Humanoid | 0.019 | 0.946 | 1.033 |
| HumanoidStandup | 0.029 | 0.940 | 1.032 |

*(b)* Value update fidelity ratio.

| Environment | Std | 1st Percentile | 99th Percentile |
|---|---|---|---|
| Ant | 0.027 | 0.892 | 1.030 |
| Humanoid | 0.023 | 0.928 | 1.051 |
| HumanoidStandup | 0.068 | 0.876 | 1.071 |

*Table 3.* Average effective update ratio. Lower is less variability.

| Method | Policy | Critic |
|---|---|---|
| Intentional update | 2.61 | 1.84 |
| No intentional scaling ($\alpha = 1$) | 1.83 | 2.99 |

current prediction toward the momentary bootstrap target. ObGD instead treats overshoot in terms of the *updated* TD error $\delta_t(\boldsymbol{w}_{t+1})$, where the target itself shifts with $\boldsymbol{w}$ through $V_{\boldsymbol{w}}(s_{t+1})$. Under a first-order approximation, the ObGD sign test depends on $\langle \nabla_{\boldsymbol{w}} \delta_t(\boldsymbol{w}_t), \boldsymbol{g}_t \rangle$ with $\nabla_{\boldsymbol{w}} \delta_t(\boldsymbol{w}_t) = \gamma \boldsymbol{g}_{t+1} - \boldsymbol{g}_t$. When $\gamma \boldsymbol{g}_{t+1} \approx \boldsymbol{g}_t$ (e.g., near self-loops or when representations are similar), this inner product can be small even if $\|\boldsymbol{g}_t\|$ is large, allowing steps that move $V_{\boldsymbol{w}}(s_t)$ far past $V_t^{\text{target}}$ without flipping the sign of $\delta_t(\boldsymbol{w})$. To avoid such cases, StreamX also introduces a bound based on a Lipschitz assumption on $\delta_t(\boldsymbol{w})$ (Elsayed et al. 2024). Overall, ObGD is naturally viewed as a safety test on a moving TD error, whereas Intentional TD directly controls the intended change in the prediction at the sampled input.

For policy learning, prior work has focused on following the *natural gradient* (e.g., see Amari 1998; Kakade 2001) which ensures consistent movement in the space of probability distributions rather than parameter space. This has led to approximations which control policy change by penalizing or constraining KL divergence between successive policies, as in trust-region and PPO-style methods (Schulman et al. 2015; 2017). Our Intentional Policy update uses a cheaper streaming proxy—targeting the one-step change in sampled log-probability—to set the typical scale of local policy change without requiring batch KL estimates.

## 9. Discussion, Limitations, and Future Work

**Intentional updates** select step sizes based on their effects on predictions and policies rather than movements in parameter space. The method specifies an intended change for a quantity of interest, then builds on intuitions from NLMS to derive a step size that achieves that change while integrating entrywise scaling, eligibility traces, and other RL-specific components. From this viewpoint, the role of

$\eta$ differs from a conventional learning rate; while still a meta-parameter, it is expressed in functional units, which clarifies its interpretation and greatly improves transferability across environments. In our experiments, a single setting was reused across benchmark families rather than tuned separately for each task. Furthermore, the method operates in an efficient streaming regime: Intentional AC uses a batch size of 1 and no replay buffer, running smoothly on a single CPU. In contrast, SAC relies on large-batch updates that typically require a GPU. In our implementation, one SAC update requires more than 100 times the FLOPs of one Intentional AC update; a detailed FLOP analysis is provided in Appendix E.

There remain several open problems and natural next steps. One open issue is the action-dependence of policy normalization. Because the action is part of the sample, action-dependent step-size rules can reweight actions inside the expectation and thereby deviate from the conventional policy-gradient direction (Appendix B). It would be preferable to obtain the same control over the scale of one-step policy change with an action-independent step size (state-dependent at most); see Appendix B for detailed discussion.

A second issue arises in both value and policy learning from shared parameters across states. Intentional updates may choose a large step size when the current gradient is small, and that update can induce large unintended changes at other states where the network is more sensitive. Appendix C discusses a conservative safeguard for this kind of cross-sample effect; it did not help in our benchmarks but may matter in other regimes; deeper exploration of which is warranted. Simple caps such as $\min(\alpha_t, 1)$ may also be useful as backstops against rare extreme steps.

Step-size meta-optimization is a natural complement to Intentional updates. We deliberately avoided learning-rate schedules and decay to respect the continual nature of RL. Intentional updates still use a global scale $\eta$, and meta step-size methods can adapt $\eta$ online, discovering effective update scales on the fly and shrinking them when recent updates are counterproductive (e.g., see Sutton 1992; Sharifnassab et al. 2025). This provides a feedback mechanism for cross-sample side effects without hand-designed schedules.

A natural extension is to use reparameterized actor updates. Our policy results use likelihood-ratio gradients; extending the same targeting idea to DDPG- (Lillicrap et al. 2016) and SAC-style (Haarnoja et al. 2018) actors will require choosing an appropriate unit of change. Finally, although our focus is streaming learning, the same unit-of-control idea may also be useful in batch and replay-based methods, potentially reducing reliance on learning-rate schedules and improving robustness across tasks.

# Acknowledgments

The authors gratefully acknowledge Joseph Modayil, Sorina Lupu, Khurram Javed, John D. Martin, and anonymous reviewers for their valuable feedback during the development of this work. We are also appreciative of the computing resources provided by the Digital Research Alliance of Canada and the financial support from the CCAI Chairs program, the RLAI laboratory, and Amii.

# Impact Statement

This paper presents work whose goal is to advance the field of Machine Learning. There are many potential societal consequences of our work, none of which we feel must be specifically highlighted here.

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

# Appendix

## A. Derivations

This appendix provides two derivations used in Sections 4 and 5: (i) the conservative trace-based targeting rule (12), and (ii) the small-step relationship in (13) that motivates using $\Delta \log \pi$ as a proxy for local policy change.

**Trace-based aggregate targeting (Eq. 12).** Recall the trace-based update (8),

$$\boldsymbol{w}_{t+1} = \boldsymbol{w}_t + \alpha_t\, \delta_t\, \boldsymbol{\rho}_t\, \boldsymbol{z}_t, \qquad \boldsymbol{z}_t = \lambda\gamma \boldsymbol{z}_{t-1} + \boldsymbol{g}_t,$$

and the aggregate targeting objective (9). Under a first-order approximation, for any $\tau \leqslant t$,

$$V_{\boldsymbol{w}_{t+1}}(s_\tau) - V_{\boldsymbol{w}_t}(s_\tau) \approx \nabla_{\boldsymbol{w}} V_{\boldsymbol{w}_t}(s_\tau)^\top (\boldsymbol{w}_{t+1} - \boldsymbol{w}_t) = \alpha_t\, \delta_t\, \langle \boldsymbol{g}_\tau, \boldsymbol{\rho}_t \boldsymbol{z}_t \rangle,$$

where $\boldsymbol{g}_\tau \doteq \nabla_{\boldsymbol{w}} V_{\boldsymbol{w}_\tau}(s_\tau)$ as in the main text. Substituting (A) into the squared form of (9) yields

$$\sum_{\tau \leqslant t} (\lambda\gamma)^{t-\tau} \Big( V_{\boldsymbol{w}_{t+1}}(s_\tau) - V_{\boldsymbol{w}_t}(s_\tau) \Big)^2 \approx \alpha_t^2\, \delta_t^2 \sum_{\tau \leqslant t} (\lambda\gamma)^{t-\tau} \langle \boldsymbol{g}_\tau, \boldsymbol{\rho}_t \boldsymbol{z}_t \rangle^2.$$

Because $\boldsymbol{\rho}_t$ is diagonal and nonnegative, it defines an inner product $\langle x, y \rangle_{\boldsymbol{\rho}_t} \doteq \langle x, \boldsymbol{\rho}_t y \rangle$. Cauchy–Schwarz in this inner product gives, for each $\tau$,

$$\langle \boldsymbol{g}_\tau, \boldsymbol{\rho}_t \boldsymbol{z}_t \rangle^2 = \langle \boldsymbol{g}_\tau, \boldsymbol{z}_t \rangle_{\boldsymbol{\rho}_t}^2 \leqslant \langle \boldsymbol{g}_\tau, \boldsymbol{g}_\tau \rangle_{\boldsymbol{\rho}_t} \langle \boldsymbol{z}_t, \boldsymbol{z}_t \rangle_{\boldsymbol{\rho}_t} = \langle \boldsymbol{g}_\tau, \boldsymbol{\rho}_t \boldsymbol{g}_\tau \rangle \langle \boldsymbol{z}_t, \boldsymbol{\rho}_t \boldsymbol{z}_t \rangle.$$

Applying (A) in (A) yields the conservative bound

$$\sum_{\tau \leqslant t} (\lambda\gamma)^{t-\tau} \Big( V_{\boldsymbol{w}_{t+1}}(s_\tau) - V_{\boldsymbol{w}_t}(s_\tau) \Big)^2 \leqslant \alpha_t^2\, \delta_t^2\, \langle \boldsymbol{z}_t, \boldsymbol{\rho}_t \boldsymbol{z}_t \rangle \sum_{\tau \leqslant t} (\lambda\gamma)^{t-\tau} \langle \boldsymbol{g}_\tau, \boldsymbol{\rho}_t \boldsymbol{g}_\tau \rangle. \tag{19}$$

Let $\sigma_\tau \doteq \langle \boldsymbol{g}_\tau, \boldsymbol{\rho}_\tau \boldsymbol{g}_\tau \rangle$ and $\bar{\sigma}_t \doteq \sum_{\tau \leqslant t} (\lambda\gamma)^{t-\tau} \sigma_\tau$ as in (11). In implementation, $\bar{\sigma}_t$ is maintained by a discounted running estimate, and $\boldsymbol{\rho}_t$ changes slowly; accordingly, we use $\bar{\sigma}_t$ as a proxy for the second sum in (19). Choosing

$$\alpha_t = \frac{\eta}{\sqrt{\bar{\sigma}_t\, \langle \boldsymbol{\rho}_t \boldsymbol{z}_t, \boldsymbol{z}_t \rangle}}$$

makes the right-hand side of (19) equal to $\eta^2 \delta_t^2$ and therefore ensures

$$\sqrt{\sum_{\tau \leqslant t} (\lambda\gamma)^{t-\tau} \Big( V_{\boldsymbol{w}_{t+1}}(s_\tau) - V_{\boldsymbol{w}_t}(s_\tau) \Big)^2} \lesssim \eta\, |\delta_t|,$$

which is the conservative form of the targeting condition (9) and yields (12).

**A small-step proxy for local KL change (Eq. 13).** Fix a state $s$ and consider a small parameter change $\Delta\boldsymbol{\theta} \doteq \boldsymbol{\theta}_{t+1} - \boldsymbol{\theta}_t$. A second-order expansion of the state-conditional KL divergence gives

$$D_{\mathrm{KL}}\big(\pi_{\boldsymbol{\theta}_t}(\cdot|s) \,\|\, \pi_{\boldsymbol{\theta}_t + \Delta\boldsymbol{\theta}}(\cdot|s)\big) \approx \frac{1}{2} \Delta\boldsymbol{\theta}^\top F(\boldsymbol{\theta}_t; s)\, \Delta\boldsymbol{\theta}, \tag{20}$$

where $F(\boldsymbol{\theta}_t; s)$ is the Fisher information matrix at $s$,

$$F(\boldsymbol{\theta}_t; s) \doteq \mathbb{E}_{a \sim \pi_{\boldsymbol{\theta}_t}(\cdot|s)} \big[ \boldsymbol{g}(a|s) \boldsymbol{g}(a|s)^\top \big], \qquad \boldsymbol{g}(a|s) \doteq \nabla_{\boldsymbol{\theta}} \log \pi_{\boldsymbol{\theta}_t}(a|s).$$

Under the same first-order approximation used in (15),

$$\Delta \log \pi(a|s) \doteq \log \pi_{\boldsymbol{\theta}_t + \Delta\boldsymbol{\theta}}(a|s) - \log \pi_{\boldsymbol{\theta}_t}(a|s) \approx \boldsymbol{g}(a|s)^\top \Delta\boldsymbol{\theta}.$$

Taking squared expectation over $a \sim \pi_{\boldsymbol{\theta}_t}(\cdot|s)$ gives

$$\mathbb{E}\Big[ \big( \Delta \log \pi(a|s) \big)^2 \Big] \approx \Delta\boldsymbol{\theta}^\top \mathbb{E}\big[ \boldsymbol{g}(a|s) \boldsymbol{g}(a|s)^\top \big] \Delta\boldsymbol{\theta} = \Delta\boldsymbol{\theta}^\top F(\boldsymbol{\theta}_t; s) \Delta\boldsymbol{\theta}. \tag{21}$$

Combining (20) and (21) yields

$$D_{\mathrm{KL}}\big(\pi_{\boldsymbol{\theta}_t}(\cdot|s) \,\|\, \pi_{\boldsymbol{\theta}_{t+1}}(\cdot|s)\big) \approx \frac{1}{2} \mathbb{E}\Big[ \big( \Delta \log \pi(a|s) \big)^2 \Big],$$

which is Eq. (13).

## B. Reweighting and Bias in Sample-Dependent Step-Sizes

In the Intentional updates, the step size is selected so that the update achieves an intended outcome for that sample. A consequence is that the expected update direction generally changes compared to using a constant step size. In this appendix, we separate two cases: (i) value learning, where sample-dependent step sizes mainly reweight states, which mainly changes how quickly different parts of the stream are tracked, and (ii) policy learning, where action-dependent step sizes can reweight actions (potentially changing the objective).

**Value learning reweights states** (usually harmless). Consider a simple case of linear TD(0), where $V_{\boldsymbol{w}}(s_t) = \boldsymbol{w}^\top \boldsymbol{x}_t$ and $\boldsymbol{x}$ is the feature representation of $s_t$. Then, under the step-size (7), the TD update (1) simplifies to

$$\boldsymbol{w}_{t+1} \;=\; \boldsymbol{w}_t + \frac{\eta}{\|\boldsymbol{x}_t\|^2}\,\delta_t\,\boldsymbol{x}_t.$$

In a stationary setting, the expectation of $\Delta \boldsymbol{w}$ would be equivalent to the expected $\Delta \boldsymbol{w}$ of a fixed step-size TD update under a different sample distribution that reweights the probability of each sample $s_t$ by $1/\|\boldsymbol{x}_t\|^2$. This is a generic consequence of sample-dependent step sizes (including NLMS): under stationarity, it can change the effective weighting over samples, and therefore can change the stationary fixed point.

In streaming RL, there is no fixed distribution and no fixed target; the goal is stable tracking as the stream changes. In that regime, this reweighting is largely the mechanism that makes each step produce a comparable change in the current predictions. Note that under large enough model capacity (e.g., in a tabular setting), with Intentional TD, the value of all samples would still converge to the true values, albeit with different rates than a fixed step-size.

**Policy learning can reweight actions** (potentially harmful). For policy gradients, the sample includes $a \sim \pi_{\boldsymbol{\theta}}(\cdot|s)$. If normalization depends on $a$, then actions are reweighted inside the expectation.

Fix a state $s$. The usual likelihood-ratio update is

$$\Delta \boldsymbol{\theta} \;=\; \alpha\, A(s,a)\, \boldsymbol{g}(s,a), \qquad \boldsymbol{g}(s,a) \doteq \nabla_{\boldsymbol{\theta}} \log \pi_{\boldsymbol{\theta}}(a|s),$$

with conditional expectation

$$\mathbb{E}[\Delta \boldsymbol{\theta} \mid s] \;=\; \alpha\, \mathbb{E}_{a \sim \pi(\cdot|s)}[A(s,a)\, \boldsymbol{g}(s,a)]. \tag{22}$$

If instead we normalize each sample update by $\|\boldsymbol{g}(s,a)\|_2^2$, then

$$\mathbb{E}[\Delta \boldsymbol{\theta} \mid s] \;=\; \alpha\, \mathbb{E}_{a \sim \pi(\cdot|s)}\left[\frac{A(s,a)}{\|\boldsymbol{g}(s,a)\|_2^2}\, \boldsymbol{g}(s,a)\right],$$

which is generally not a constant multiple of (22); it is rather applying policy gradient update on the scaled advantage $A(s,a)/\|\boldsymbol{g}(s,a)\|_2^2$. The normalization has made the effective advantage action-dependent. This can change the fixed point for each sample, even in the tabular setting. A simple failure mode is possible with two actions: it is possible that $A(s,a_1) > A(s,a_2) > 0$ while $A(s,a_1)/\|\boldsymbol{g}(s,a_1)\|_2^2 < A(s,a_2)/\|\boldsymbol{g}(s,a_2)\|_2^2$, in which case the normalized update can favor $a_2$ even though $a_1$ is better.

To prevent such bias, we need a step-size $\alpha_t$ that is action-independent, while it may still be a function of the current state. Such methods need further exploration in future work. Empirically, the action dependence of the normalizers we use in Algorithm 3 did not prevent good performance in our experiments.

## C. Cross-Sample Effects and Conservative Safeguards

Intentional update chooses $\alpha_t$ to enforce a desired change on the current sample. With shared parameters, a step that is appropriate for the current sample can occasionally cause large unintended changes elsewhere. This appendix makes the mechanism explicit and summarizes a simple conservative safeguard.

**Why collateral change can occur.** For TD(0) with $\Delta \boldsymbol{w}_t = \boldsymbol{w}_{t+1} - \boldsymbol{w}_t = \alpha_t \delta_t\, \boldsymbol{g}_t$ and $\alpha_t = \eta/\|\boldsymbol{g}_t\|_2^2$ from (7),

$$\|\Delta \boldsymbol{w}_t\|_2 = \alpha_t\, |\delta_t|\, \|\boldsymbol{g}_t\|_2 = \eta\, \frac{|\delta_t|}{\|\boldsymbol{g}_t\|_2}.$$

Thus, when the current sensitivity $\|\boldsymbol{g}_t\|_2$ is unusually small, achieving a fixed amount of prediction change at $s_t$ can require a large parameter step. At any other state $s$, the induced prediction change satisfies

$$\left|V_{\boldsymbol{w}_{t+1}}(s) - V_{\boldsymbol{w}_t}(s)\right| \approx \left|\nabla_{\boldsymbol{w}} V_{\boldsymbol{w}_t}(s)^\top \Delta \boldsymbol{w}_t\right| \leqslant \|\nabla_{\boldsymbol{w}} V_{\boldsymbol{w}_t}(s)\|_2 \|\Delta \boldsymbol{w}_t\|_2.$$

The same argument applies to action values and to policy outputs: the current sample determines $\alpha_t$, while other inputs experience the resulting parameter change.

**A two-scale denominator.** One conservative guard is to prevent the denominator from becoming much smaller than its typical value. Define $u_t \doteq \|\boldsymbol{g}_t\|_2^2$ and maintain an exponential average $\bar{u}_t$. Use the guarded step size

$$\alpha_t = \frac{\eta}{\max\left(u_t, \sqrt{u_t \bar{u}_t}\right)}. \tag{23}$$

When $u_t$ is typical, (23) reduces to (7). When $u_t$ is unusually small, the denominator becomes $\sqrt{u_t \bar{u}_t}$, which limits the parameter-step magnitude by a typical sensitivity scale:

$$\|\Delta \boldsymbol{w}_t\|_2 = \alpha_t |\delta_t| \|\boldsymbol{g}_t\|_2 \leqslant \eta \frac{|\delta_t|}{\sqrt{\bar{u}_t}}.$$

An analogous guard can be applied to the trace-based denominator $d_t \doteq \sqrt{\bar{\sigma}_t \langle \boldsymbol{\rho}_t \boldsymbol{z}_t, \boldsymbol{z}_t \rangle}$ from (12) by replacing $u_t$ with $d_t$ and $\bar{u}_t$ with a running average $\bar{d}_t$.

**Notes.** In our MuJoCo experiments, these conservative guards did not improve performance, but they isolate a real mechanism by which sample-wise targeting can produce collateral change. A different way to respond is to adapt the global scale $\eta$ online (e.g., by meta step-size methods): when recent updates make current learning harder, the effective step size is reduced.

# D. Experiment Details

We follow the benchmark suites, implementation style, and evaluation protocol of (Elsayed et al. 2024). All experiments are fully streaming: each environment interaction yields at most one learning update, with no replay buffer, no mini-batches, and no multiple passes over stored experience. We implement all agents in Python using PyTorch for automatic differentiation and Gymnasium for environment interfaces, matching the reference setup (Elsayed et al. 2024).

For time-limited episodic tasks, we use a continual approximation by bootstrapping at truncation (timeouts) rather than treating timeouts as true terminals. Unless stated otherwise, curves show the mean over 30 independent runs with 95% confidence intervals.

## D.1. Continuous control: MuJoCo Gym and DM Control Suite

We train a streaming actor–critic (*Intentional AC*) where the critic uses Intentional TD($\lambda$) (Algorithm 1) and the actor uses Intentional Policy Gradient (Algorithm 3). We use the same MuJoCo Gym and DM Control task suites as Elsayed et al. (2024), and each run consists of 5 million environment steps.

Both the policy and value functions are parameterized by separate fully connected networks with two hidden layers of width 128 (i.e., 128×128), using LeakyReLU nonlinearities, and LayerNorm applied before each activation. The policy network uses two output heads: one for the action mean and one for the action standard deviation. For continuous actions, the standard deviation is parameterized via SoftPlus, $f(x) = \log(1 + e^x)$, and for numerical stability, we switch to the linear mapping $y = x$ when the input exceeds 20. Actions are clamped to the range $[-1, 1]$. (For discrete policies, the reference protocol uses a softmax parameterization; our discrete-control experiments below use Q-learning with $\epsilon$-greedy behavior.)

For Intentional AC, we used $\eta = 0.5$ for critic and $\eta = 0.05$ for policy, which translates into roughly 5 percent change in action-probability after each update, on average. These meta-parameters were found for MuJoCo with coarse sweeping $\eta_{\text{critic}} \in \{0.5, 1/3\}$ and $\eta_{\text{actor}} \in \{0.05, 1/30\}$, and then used also for DMC suite experiments with no further sweeping. We set $\beta_{\text{clip}} = \beta_{\text{norm}} = 0.9998$ without any sweeping, based on the intuition to obtain averages over roughly 5000 past recent steps. We keep the discount and trace settings used in the reference protocol, $\gamma = 0.99$ and $\lambda = 0.8$. In (Elsayed et al.

2024), StreamAC uses ObGD with step size $\alpha = 1$, with $\kappa = 3$ for the policy network and $\kappa = 2$ for the value network, and SparseInit with sparsity ratio $s = 90\%$; we keep the same architectural and initialization choices and only change the learning update to the intentional-update rules unless an ablation explicitly removes a component (Elsayed et al. 2024). For batch baselines reported in our experiments, PPO and SAC are taken from the CleanRL implementation (Huang et al. 2022).

### D.2. Discrete-action control: MinAtar

We evaluate Intentional Q-learning on the MinAtar suite under the same fully streaming constraints (one update per step, no replay) as (Elsayed et al. 2024). Each run is trained for 5M environment steps. The Q-network matches the reference architecture: one convolutional layer with 16 filters of size $3 \times 3$ and stride 1, followed by flattening to 1024 features, then one fully connected hidden layer with 128 units. We use LeakyReLU activations with LayerNorm inserted before each activation. We use $\gamma = 0.99$ and $\lambda = 0.8$ as in the reference setup. For Intentional Q($\lambda$), we use $\eta = 0.25$, which we found by sweeping over $\eta \in \{0.5, 0.25, 1/6\}$. Similar to Intentional AC, we set $\beta_{\text{clip}} = 0.9998$ with no sweeping. In the reference StreamX implementation, ObGD uses $\alpha = 1$ and $\kappa = 2$, and SparseInit uses sparsity ratio $s = 90\%$; we keep the same network/initialization conventions and only replace the update with our intentional-update rule unless otherwise stated. (Elsayed et al. 2024). Exploration is $\epsilon$-greedy with a linear schedule from $\epsilon = 1$ to $\epsilon = 0.01$, reaching $\epsilon = 0.01$ after 5% of total training steps. We use the DQN implementation given in CleanRL.

### D.3. Discrete-action control: Atari

We evaluate Intentional Q-learning on Atari under the streaming setup of Elsayed et al. (2024). Each run uses 50M action steps, corresponding to 200M frames in total, with each selected action repeated 4 times. We use $\gamma = 0.99$ and $\lambda = 0.8$, and $\epsilon$-greedy exploration with the same linear schedule as MinAtar (from 1 to 0.01, reaching 0.01 at 5% of training).

The Q-network matches the reference architecture: three convolutional layers (32 filters of $8 \times 8$ with stride 5; 64 filters of $4 \times 4$ with stride 3; 64 filters of $3 \times 3$ with stride 2), followed by one fully connected hidden layer with 256 units and a linear output layer over actions. We use LeakyReLU nonlinearities with LayerNorm placed before each activation. For Intentional Q($\lambda$), we use $\eta = 0.25$, which we found by sweeping over $\eta \in \{0.25, 1/6\}$. Similar to the MinAtar and Mujoco experiments, we set $\beta_{\text{clip}} = 0.9998$ with no sweeping. In the reference StreamX implementation, ObGD uses $\alpha = 1$ and $\kappa = 2$, and SparseInit uses sparsity ratio $s = 90\%$; we keep the same network/initialization conventions and only replace the update with our intentional-update rule unless otherwise stated (Elsayed et al. 2024).

The Atari environment protocol follows Hessel et al. (2018) with specific changes that make the setting more challenging, as in Elsayed et al. (2024). Frames are downsampled to $84 \times 84$ and converted to grayscale; to mitigate partial observability, we stack 4 consecutive frames. At episode start, we apply a random number of no-op actions (up to 30). For games that remain static until a firing action is used, we also take a random initial action accordingly when applicable. Episodes are terminated on loss of life. Unlike Hessel et al. (2018), we do not clip rewards and we do not scale pixel values by dividing by 255 (Elsayed et al. 2024). The DQN scores at 200M frames used in the reference Atari plots are taken from Table 6 of Hessel et al. (2018).

### D.4. Value prediction under a fixed policy

To isolate prediction from control, we evaluate streaming value learning under a fixed behavior policy. For each MuJoCo task, we first train an actor using StreamAC (Elsayed et al. 2024) for 5M environment steps, then freeze the policy and train a critic from scratch on the resulting stream. We measure performance by the root mean squared prediction error $\sqrt{\mathbb{E}_s\left[(V_{\mathbf{w}_t}(s) - G(s))^2\right]}$, where $G(s)$ denotes a Monte Carlo estimate of the discounted return from $s$ under the fixed policy (estimated from rollouts), and the expectation is over states visited by that policy.

### D.5. Ablations

We run two ablation suites. First, we ablate components of the intentional update by removing diagonal normalization, and $\delta$-clipping individually while keeping all other settings fixed. Second, we follow the StreamX-style component-removal protocol of Elsayed et al. (2024) and remove reward scaling, observation normalization, and sparse initialization while keeping the intentional update unchanged, to test whether intentional updates reduce dependence on these auxiliary stabilizers in the streaming regime.

# E. Approximate Per-Update FLOPs Comparison with SAC

This appendix gives a rough comparison between the cost of one Intentional AC update and one SAC update. The purpose is not to provide an exact hardware benchmark, but to make the scale difference between the two methods more explicit. The comparison is stated in terms of approximate floating-point work under a simple large-network model, ignoring scalar overheads and other lower-order costs.

We write $N$ for the number of parameters in one network, and assume that the actor and critic networks are of roughly similar size. We also assume sufficiently large feedforward networks so that the dominant cost is linear in $N$.

**Intentional AC.**  Intentional AC uses two networks, one actor and one critic, and operates in a streaming regime with batch size 1 and no replay. For each network, one update consists of:

- forward pass: $2N$,
- backward pass: $4N$,
- RMSProp scaling ($\rho_t$): $5N$,
- computation of gradient norm $\langle \rho_t \boldsymbol{g}_t, \boldsymbol{g}_t \rangle$: $3N$,
- eligibility trace $\boldsymbol{z}_t$ update: $2N$,
- computation of trace norm $\langle \rho_t \boldsymbol{z}_t, \boldsymbol{z}_t \rangle$: $3N$,
- parameter update: $3N$.

Thus, the total cost per network is

$$2N + 4N + 5N + 3N + 2N + 3N + 3N = 22N.$$

Since Intentional AC uses two networks, the total per-update cost is

$$2 \times 22N + 2N = 46N,$$

where the additional $2N$ is needed for constructing the TD target.

**SAC.**  SAC uses five networks in total: one policy network and four critic networks. It also uses replay with batch size 256. For each network, gradient computation consists of:

- forward pass: $2 \times 256\,N$,
- backward pass: $4 \times 256\,N$,
- batch aggregation: $1 \times 256\,N$.

This gives $(2 + 4 + 1) \times 256\,N = 1792N$. The Adam update then adds:

- RMS term update: $5N$,
- momentum update: $3N$,
- parameter update: $3N$,

for an additional $5N + 3N + 3N = 11N$. Thus, the total per-network cost is

$$1792N + 11N = 1803N.$$

Three out of five networks are updated with this much compute per step, and each of the other two target networks needs a forward pass over the batch to construct the TD target. We ignore the infrequent checkpointing of the target networks. Therefore, the total per-update cost is

$$3 \times 1803N + 2 \times 256 \times 2N = 6433N.$$

**Resulting ratio.** Under these assumptions, the ratio between one SAC update and one Intentional AC update is

$$\frac{6433N}{46N} \approx 140.$$

Thus, one SAC update requires more than $100\times$ the floating-point work of one Intentional AC update.

**Interpretation.** This comparison is only approximate, but it helps clarify the computational regimes of the two methods. Intentional AC is designed for cheap streaming updates, with batch size 1, no replay, and modest per-step computation. SAC, by contrast, relies on large-batch replay updates over multiple networks and is correspondingly much more expensive per update. In practice, this difference is large enough that Intentional AC is comfortable to run on CPU and may be of interest in resource-constrained settings such as edge devices, whereas SAC is more naturally studied in a GPU regime. A compute-matched performance comparison based on these FLOPs estimates is provided in Appendix F.2.

# F. Further Experiment Results

## F.1. Empirical Analysis of Action-Dependent Normalization Bias

In this section, we test the action-dependent normalization bias, discussed in Appendix B and references at the end of Section 5. To quantify the impact of action-dependent scaling, we evaluate the cosine similarity between the expected Intentional update and the unbiased update ($\alpha = 1$). We analyzed Intentional AC across 30 seeds at the 1M environment step mark, corresponding to the mid-training "rising phase". For each seed, we sampled 1,000 states, and for each state, we estimated the expected update direction by averaging over 1,000 sampled actions. We report statistics over these 30,000 total samples per environment. We used $\lambda = 0.8$ for the training phase, and $\lambda = 0$ for the bias-measurement phase.

The results, detailed in Table 4, indicate high alignment in the Humanoid environments (median $\approx 0.96$), suggesting minimal bias during the critical learning phases where high-quality update directions are most necessary. In contrast, *Ant-v4* exhibits lower alignment. The policy-bias here is neither negligible nor overwhelming, and resolving it may further improve performance, marking a promising direction for future research, as outlined in Section 9.

*Table 4.* Cosine similarity between the expected Intentional update and the unbiased update (closer to 1 means less bias).

| Environment | Mean | Std | Median | 20th Percentile | 5th Percentile | 1st Percentile |
|---|---|---|---|---|---|---|
| Ant-v4 | 0.590 | 0.302 | 0.628 | 0.342 | 0.025 | -0.261 |
| Humanoid-v4 | 0.930 | 0.090 | 0.963 | 0.893 | 0.754 | 0.559 |
| HumanoidStandup-v4 | 0.904 | 0.107 | 0.944 | 0.845 | 0.685 | 0.500 |

## F.2. Compute-Matched Performance Comparison

In this experiment, we compared the final average return of Intentional AC at 5M environment steps against SAC evaluated at the specific number of steps that yields the identical total update compute budget, based on the FLOPs estimates of Appendix E.

As shown in Table 5, Intentional AC maintains a substantial performance advantage under matched compute constraints. This demonstrates that Intentional AC achieves higher performance per unit of compute, indicating a more effective utilization of the available computational budget in streaming or strictly constrained RL setups.

## F.3. Are Supervised Normalization Techniques Sufficient

In this experiment, we explored whether standard supervised-learning normalization techniques alone are sufficient for meaningful learning in a streaming setting. We compared Intentional AC against standard supervised-learning normalization techniques and two alternative baselines in the fully streaming setting. Given a typical RL update vector $g_t = \delta_t z_t$, where $z_t$ is the eligibility trace of $\nabla V$ gradient, or $\nabla \log \pi$, we evaluate:

1. *Standard-RMSProp:* The raw update $g_t$ is fed directly into an RMSProp optimizer, representing the dominant approach in deep RL.

*Table 5.* Average return under matched compute budgets. Intentional AC is evaluated at 5M steps, while SAC is evaluated at the step count that equates to the same total FLOPs. We report the mean return $\pm$ standard error.

| Environment | Intentional AC (5M steps) | SAC (compute-matched steps) |
|---|---|---|
| Ant | $\textbf{5513}_{\pm 54}$ | $144_{\pm 32}$ |
| HalfCheetah | $\textbf{5064}_{\pm 288}$ | $1383_{\pm 91}$ |
| Hopper | $\textbf{2254}_{\pm 71}$ | $302_{\pm 16}$ |
| Humanoid | $\textbf{6227}_{\pm 80}$ | $402_{\pm 13}$ |
| HumanoidStandup | $\textbf{189k}_{\pm 5.1k}$ | $89k_{\pm 5.1k}$ |
| Walker2d | $\textbf{3515}_{\pm 113}$ | $257_{\pm 22}$ |

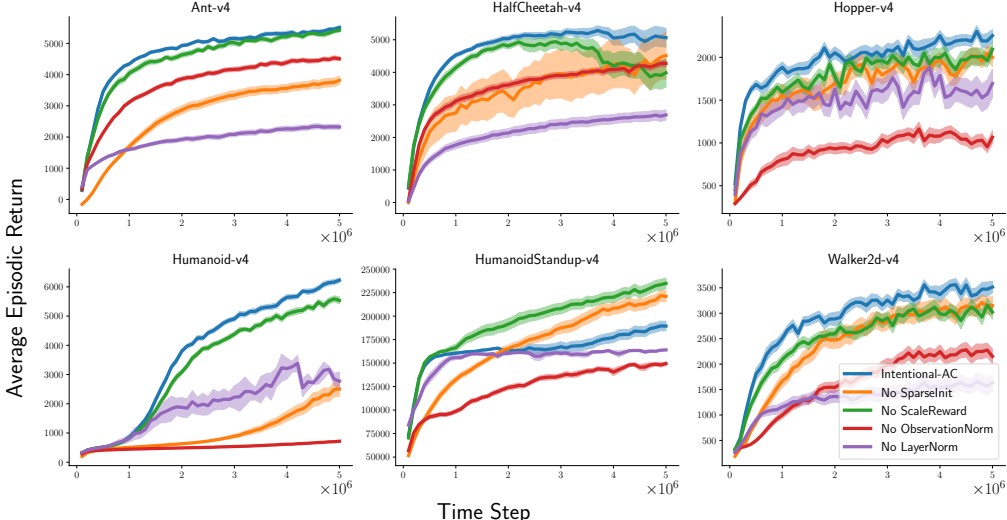

*Figure 7.* Ablation: Robustness to StreamX stabilizers.

2. *Normalized-Gradient Baseline:* To emulate supervised-learning-style normalization, we divide the Standard-RMSProp update by its own squared norm ($\|g_t\|^2$). This enforces a fixed-magnitude change but lacks the specific functional unit decomposition of the intentional-update approach.

We swept over $\eta \in \{10^{-1}, 10^{-2}, 10^{-3}, 10^{-4}, 10^{-5}\}$ and ran 10 seeds per environment. As demonstrated in Table 6, neither baseline showed meaningful learning in the MuJoCo environments. These catastrophic failures align with the "streaming barrier" described by Elsayed et al. (2024), where conventional methods reliant on batching or replay buffers collapse in purely streaming contexts. The success of Intentional AC highlights the necessity of its principled, functional normalization over naive gradient scaling.

*Table 6.* Performance comparison of Intentional AC against standard gradient and normalized-gradient baselines in a streaming setting.

| Environment | RMSProp | RMSProp + Norm. | Intentional AC |
|---|---|---|---|
| Ant-v4 | -250 | -300 | **5513** |
| Humanoid-v4 | 350 | 7 | **6227** |
| HumanoidStandup-v4 | 59.5k | 3.8k | **189k** |

## F.4. Further Ablation Results

Here we provide complementary results for the ablation study in Section 7.6. Fig. 7 plots the returns throughout the learning process in the absence of different StreamX preconditioners. Fig. 8 provides bar diagrams for final returns of Intentional AC and StreamX, as an alternative visualization of Table 1.

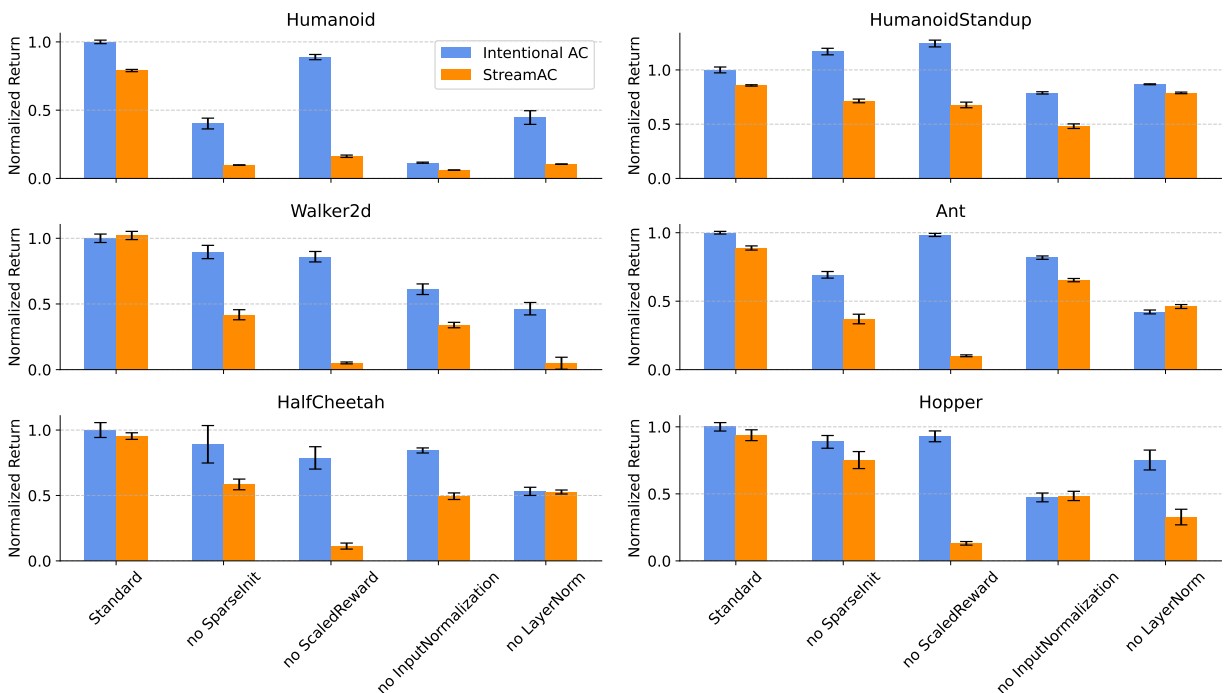

*Figure 8.* Robustness to StreamX stabilizers.

