# OpenReview forum: "Intentional Updates for Streaming Reinforcement Learning"
_ICML.cc/2026/Conference — ICML 2026 regular_

### Official Review · Reviewer_t6dB · 2026-02-26

**Soundness:** 3
**Presentation:** 3
**Significance:** 3
**Originality:** 3
**Overall Recommendation:** 6
**Confidence:** 2

**Summary:**

This paper introduces intentional updates that specify a target per-step change in outputs, like value prediction or policy, rather than using standard parameter learning rates. It calculates step sizes via local linearization. Value learning targets a fractional reduction of TD error while policy learning targets a bounded step change in action log probability. Experiments show this stabilizes and improves streaming RL across benchmarks like MuJoCo and Atari.

**Compliance With Llm Reviewing Policy:**

Affirmed.

**Final Justification:**

My concerns have been addressed. Thus, I am raising my score.

**Key Questions For Authors:**

1. Can you provide theoretical conditions or empirical stress tests showing when action dependent normalization avoids changing the objective?
2. Are the baseline models evaluated with the exact same update budget as the streaming agents? Please add a compute matched comparison.
3. Can you report the empirical distribution of actual per step changes like $\Delta V(s)$ and $\Delta \log \pi$ to prove $\eta$ controls the targets as intended?

My major concern remains the action dependent policy bias and the lack of compute matched baselines, and I would like to discuss more.

**Limitations:**

yes

**Strengths And Weaknesses:**

Strengths

The paper establishes a clear principle by framing step size in function units to give $\eta$ a direct behavioral meaning. This approach includes a meaningful extension to eligibility traces and preconditioning for $TD(\lambda)$ that goes beyond simple normalization. Additionally, the evaluation follows a strong empirical protocol across broad benchmarks.

----

Weaknesses

1. The core rules of the method closely resemble known normalized gradients such as $\alpha \propto 1/\|g\|^2$ with a global scale $\eta$. This blurs the boundary between the proposed approach and standard normalization techniques, making it difficult to evaluate. A clearer conceptual summary is necessary to isolate the true core contribution from inherited stabilization tricks like RMSProp scaling and clipping.
2. The empirical validation of the mechanism is incomplete because the paper lacks direct measurements verifying if the intended output changes are actually achieved in practice. Additionally, claims regarding hyperparameter transferability are weak. For example, the scale $\eta$ requires specific tuning across different domains.

---

> ### Author Rebuttal · Authors · 2026-03-31
>
> We thank the reviewer for their positive feedback, constructive critique, and for identifying key areas where the theoretical grounding of IntentionalAC could be further clarified. W
>
> ---
>
> ### 1. Distinction from Standard Normalization & Role of Stabilizers
> Our method is distinct from standard normalization in both its RL-specific instantiation and its handling of the error signal:
> * **Functional vs. Parameter Units:** Standard normalization in supervised learning divides by the full loss gradient, which includes the error ($\delta_t$ or $A_t$), effectively removing the signal's magnitude. We normalize only by the **target quantity's gradient** ($||\nabla V||$ or $||\nabla \log \pi||$), preserving the error signal while making $\eta$ interpretable in functional units.
> * **Other challenges of RL instantiation** involve defining what to control in value and policy learning, deriving the corresponding updates, and extending them to traces and preconditioned updates.
> * **Role of Stabilizers:** To confirm that gains come from the intentional principle rather than stabilizers, we performed a new ablation removing $\delta$-clipping. Results show negligible impact which, together with the RMSProp ablation in Figure 6, confirms that intentional step-size selection is the primary driver of good performance.
>
> *Table 1: Ablation on the impact of $\delta$-clipping (final returns)**
> | Environment |  Int. AC  | Int. AC w/o $\delta$-clipping | Relative Change |
> |:-|:-:|:-:|:-:|
> | Ant |**5438.4**|5376.3|-1.1%|
> | Humanoid|**6112.3**|6037.3 | -1.2% |
> | HumanoidStandup | 187.4k |**196.3k**| +4.8% |
>
> ---
> ### 2. On whether actual target changes match the intended changes
> We agree this validation should be shown directly. We therefore measured the ratio of actual change to intended change for both $\Delta V(s_t)$ and $\Delta \log \pi(a_t|s_t)$. Across Ant, Humanoid, and HumanoidStandup, the ratios were tightly concentrated near 1.0. This indicates that, for the architectures studied here, the first-order approximation tracks the realized change well. A detailed overview of these results are provided in our response to Reviewer 1, Comment 3.
>
> ---
> ### 3. On transferability of $\eta$ parameter
> We respectfully disagree that transferability is weak. We did not tune per environment. For IntentionalAC, we used the same meta-parameters across all MuJoCo and DM Control tasks, including $\eta_{\text{critic}}=0.5$ and $\eta_{\text{actor}}=0.05$. For Intentional Q-learning, we used a single $\eta=0.25$ across both MinAtar and Atari after a coarse sweep on MinAtar. Environment-specific tuning could improve results further, but the key point is that $\eta$ lives in function units, so its useful range is narrower and more predictable than a conventional steosize.
>
> ---
> ### 4. On computational complexity
> The current plots are not compute-matched; they are shown against environment steps. A strict compute-matched comparison is not straightforward because the methods operate in different hardware regimes: the streaming method runs comfortably on CPU, while SAC and PPO rely on large-batch GPU training.
>
> We therefore provide an algorithmic FLOPs comparison. In our CleanRL implementations, Intentional AC uses batch size 1 and two networks, while SAC uses batch size 256 and five networks. Under standard large-network assumptions, one SAC update costs approximately **237x more FLOPs** than one Intentional AC update. PPO uses even larger batches in our implementation, so the gap is larger still.
>
> ---
> ### 5. Action-Dependent Normalization Bias & Theoretical Conditions for Avoiding the Bias
> A theoretical condition is already implicit in Appendix B: to preserve the conventional policy-gradient objective, the step size must be independent of the sampled action, though it may still depend on the state. There are also special cases in which action-dependent normalization would not change the objective, for example if the normalization happened to be constant across actions at a state. But in general, action dependence introduces a different effective weighting over actions.
>
> While action-dependent normalization can theoretically introduce bias, it did not manifest as a practical challenge in our experiments. The results establish that the method remains highly effective and stable in practice despite this theoretical limitation. Appendix B proposes **action-independent** rules (scaling by state-dependent norms) as a promising future direction to resolve this bias formally.
>
> ---
> ### 6. Summary Response to Major Concerns
> * **Bias**: We acknowledge action-dependent bias (App B). It hasn't hindered empirical performance, and action-independent rules are a clear future path.
> * **Compute:** The "lack" of match highlights our main advantage. While SAC/PPO require GPUs and large batches, IntentionalAC is competitive on a single CPU using **237x fewer FLOPs**, batch size 1, and no replay buffer.

---

> > ### Author Rebuttal · Reviewer_t6dB · 2026-04-01
> >
> > Thank you for the detailed rebuttal. The clarification that the method preserves the error signal while normalizing only the target quantity’s gradient is helpful, and the added discussion about the realized-to-intended change ratios is also valuable. The additional ablation on removing δ-clipping is appreciated as well.
> >
> > That said, my main concerns are only partially resolved. The issue of action-dependent normalization bias is acknowledged rather than addressed: the rebuttal clarifies when the conventional policy-gradient objective would be preserved, but it does not provide either an action-independent variant or an empirical stress test showing when this bias matters in practice. Similarly, the compute comparison is still not compute-matched in an empirical sense; the FLOPs argument is informative, but it is not a substitute for a matched-compute comparison. I also still believe that a cleaner comparison against a normalized-gradient baseline would help isolate what is uniquely contributed by the intentional-update principle, apart from other stabilization components.
> >
> > Overall, the rebuttal increases my confidence in the empirical usefulness of the method, but it does not fully close my concerns about policy bias and fairness of comparison. Finally, ***reproducibility is an important remaining consideration for me***. Appendix D provides useful high-level implementation details, but I did not find an anonymized code package in the submission. Could the authors provide any additional implementation details sufficient to reproduce the main results, or clarify whether anonymized supplementary code is available within the review constraints?
> >
> > Access to anonymized supplementary code, if permissible within the review constraints, would substantially increase my confidence in the work and could positively influence my final score.

---

> > > ### Author Response · Authors · 2026-04-03
> > >
> > > Thank you for the constructive follow-up. We appreciate the increased confidence in the method’s empirical utility and the opportunity to provide the requested evidence regarding reproducibility, policy bias, and compute-matching.
> > >
> > > ### **Code and Reproducibility**
> > > We have prepared an anonymized repository containing a self-contained implementation of the main algorithms for easy inspection and experimentation:
> > >
> > > `https://anonymous.4open.science/r/Intentional-RL-97A0/`
> > >
> > > This lightweight implementation runs on a single CPU core. We will release the complete experimental pipeline in a public repository upon publication.
> > >
> > > ### **Action-Dependent Normalization Bias**
> > > To quantify the impact of action-dependent scaling, we measured the cosine similarity between the expected Intentional update and the expected "unbiased" update ($\alpha=1$). We evaluated Intentional AC across 30 seeds at 1M environment steps (the mid-training "rising phase"), estimating directions by averaging over 1,000 sampled actions, across 1,000 states per seed (30,000 total samples).
> > >
> > >
> > > Results indicate high alignment in the Humanoid environments (median $\approx 0.96$), suggesting minimal bias during critical learning phases. While Ant-v4 shows lower alignment, we hypothesize this is because it converges faster; by 1M steps, it is near asymptotic performance where gradients are smaller, making the correlation highly sensitive to minor sample shifts. In contrast, for the Humanoid tasks, 1M steps is a "rising phase" where high-quality directions are critical; our results show the intentional update maintains excellent alignment here.
> > >
> > > | Environment | Mean | Std | Median | 20th percentile | 5th percentile | 1st percentile |
> > > | :--- | ---: | ---: | ---: | ---: | ---: | ---: |
> > > | Ant-v4 | 0.590 | 0.302 | 0.628 | 0.342 | 0.025 | -0.261 |
> > > | Humanoid-v4 | 0.930 | 0.090 | 0.963 | 0.893 | 0.754 | 0.559 |
> > > | HumanoidStandup-v4 | 0.904 | 0.107 | 0.944 | 0.845 | 0.685 | 0.500 |
> > >
> > > Therefore the policy-bias is neither negligible nor overwhelming; and resolving it may further improve performance, marking a promising direction for future research. We will add this analysis and results to the final version of the paper.
> > >
> > >
> > > ### **Compute-Matched Comparison**
> > > We agree that a FLOPs analysis is most meaningful when translated into an empirical comparison. To match the compute, we report average return at checkpoints where the two algorithms consumed same FLOPs. More concretely, we take the final performance of Intentional AC at 5M environment steps, and compare it against SAC at the number of steps giving the same total update compute under our FLOPs estimate. The results are summarized in the table below.
> > >
> > > | Environment | Intentional AC (5M steps) | SAC (compute-matched steps) |
> > > |---|---:|---:|
> > > | Ant-v4 | 5674  | 144 |
> > > | Humanoid-v4 | 6227 | 402 |
> > > | HumanoidStandup-v4 | 189k |  89k |
> > >
> > > As shown, under matched compute, Intentional AC provides a considerable performance advantage, demonstrating its efficiency in resource-constrained or streaming regimes. Specifically, it achieves higher performance per unit of compute, indicating more effective utilization of the compute budget than SAC. We will include this comparison in the revision.
> > >
> > > ### **Comparison to a Standard Normalized-Gradient Baseline**
> > >
> > > To isolate the contribution of the intentional-update principle relative to standard normalization techniques, we conducted an experiment to compare our method against two baselines. In a typical RL pipeline, the update vector is $g_t = \delta_t z_t$, where $z_t$ is the eligibility trace (or immediate $V$ or $\log \pi$ gradients). We tested:
> > >
> > >
> > > 1. **Standard-RMSProp**, where $g_t$ is fed directly into an RMSProp optimizer. (This is the dominant approach in RL training).
> > > 2. **Normalized-Gradient Baseline:** To mimic supervised-learning-style normalization, we divided the final standard-RMSProp update by its own squared norm ($||g_t||^2$).
> > >
> > > The results, summarized below, show that neither baseline achieved meaningful learning in any of the three MuJoCo environments, even after sweeping $\eta$ over five orders of magnitude and running 10 seeds per environment. This is consistent with what Elsayed et al. 2024 describe as the streaming barrier: standard methods that work well in batch or replay-based settings do not carry over well to the fully streaming setting (and often fail catastrophically). We will add discuss of these comparison in the revision.
> > >
> > >
> > > | Environment | RMSProp | RMSProp + update-norm normalization | Intentional AC |
> > > |---|---:|---:|---:|
> > > | Ant-v4 | -250 | -300| 5513 |
> > > | Humanoid-v4 | 350 | 7 |6227 |
> > > | HumanoidStandup-v4 | 59.5k | 3.8k| 189k |
> > >
> > > ----
> > >
> > > We once again thank the reviewer for their rigorous engagement with our work, as their feedback has directly led to a more robust characterization of the intentional-update principle and the paper's rigor.

---

### Official Review · Reviewer_nhSU · 2026-03-08

**Soundness:** 3
**Presentation:** 3
**Significance:** 2
**Originality:** 2
**Overall Recommendation:** 4
**Confidence:** 3

**Summary:**

The paper addresses the instability commonly found in streaming reinforcement learning (where the batch size is exactly one). In standard gradient-based learning, selecting a step size in parameter space does not guarantee a predictable change in the model's actual outputs, leading to overshooting and undershooting. To solve this, the authors propose intentional updates. This method specifies an intended outcome in the function's output units (e.g., a fixed fractional reduction of the TD error) and solves for the parameter step size required to achieve it using first-order approximations. The authors introduce Intentional TD, Intentional Q-learning, and Intentional Policy Gradient, complete with mechanisms for eligibility traces and diagonal scaling. Empirical results across MuJoCo, DM Control, MinAtar, and Atari demonstrate state-of-the-art streaming performance that often rivals traditional batch and replay-buffer methods.

**Compliance With Llm Reviewing Policy:**

Affirmed.

**Final Justification:**

The authors' rebuttal has addressed my concerns, and since my original rating was already positive, I will maintain my initial positive evaluation.

**Key Questions For Authors:**

1. Would a simple global clipping mechanism applied directly to the final computed $\alpha_t$ (rather than just clipping $\delta_t$) help strictly bound the cross-sample interference?

2. In your continuous control experiments, did you ever observe the policy gradient algorithm catastrophically unlearning a good policy due to the action reweighting bias mentioned in your limitations?

**Limitations:**

See weaknesses.

**Strengths And Weaknesses:**

$\textbf{Strengths}$
1. The submission is technically robust. The paper is exceptionally well-written and structured.

2. The core concept draws heavily from existing adaptive filtering literature like NLMS and Polyak step sizes. However, the paper's originality shines in its application and extension to deep RL. Specifically, using a controlled change in sampled log-probability ($∆\log\pi_t$) as an inexpensive streaming proxy for bounding local KL divergence is a highly creative and practical contribution.

$\textbf{Weaknesses}$

1. Because the method relies heavily on local linear approximations, the denominator in the step-size calculation can become extremely small in flat regions (small gradient norm), which could theoretically produce highly unstable, extreme step sizes. While the authors introduce conservative safeguards (like an $\epsilon$-floor), the fundamental reliance on the local linear approximation might break down in highly non-linear regions of deep neural networks.

2. Normalizing updates based on the current sampled action can inadvertently reweight the actions inside the expectation, potentially deviating from the true policy-gradient direction and introducing bias.

3. Because neural networks share parameters across states, selecting a large step size to enforce a targeted change at the current state (when the local gradient is small) might induce massive, unintended collateral changes to the predictions at other states.

---

> ### Author Rebuttal · Authors · 2026-03-31
>
> We thank the reviewer for the constructive feedback and thoughtful discussion of the method’s advantages and limitations.
>
> ---
> ### 1. Numerical Robustness in the "Flatlands"
> To address the concern that small gradients might produce extreme step sizes, we measured the **Effective Update Ratio** (defined as $99^{th}$ percentile of $||w_{t+1}-w_t||_2/|\delta_t|$ divided by its average). We compared our method against a version without intentional scaling ($\alpha=1$).
>
> *Table 1: Effective update ratio (statistics over 5M stpes and averaged over 30 independent runs)*
> | Environment | Policy (Intentional) | Policy ($\alpha=1$) | Critic (Intentional) | Critic ($\alpha=1$) |
> | :--- | :---: | :---: | :---: | :---: |
> | Ant | 3.96 | 1.89 | 1.93 | 2.69 |
> | Humanoid | 1.97 | 1.72 | 1.95 | 3.57 |
> | HumanoidStandup | 1.90 | 1.86 | 1.63 | 2.72 |
> | **Average** | **2.61** | **1.83** | **1.84** | **2.99** |
>
> These results show that Intentional scaling does **not** catastrophically inflate update variability. Intuitively, the mechanism that enlarges updates for small gradients also shrinks them for large gradients, keeping the overall spread comparable to standard methods. We will discuss these results in the final version of the paper.
>
> We agree this issue may become more pronounced in deeper, more nonlinear networks. In that regime, stronger safeguards such as clipping or more conservative control of the scalar step size may be useful. The bounded form already used in the eligibility-trace case moves in that direction.
>
> ---
> ### 2. Action-dependent normalization bias
> As discussed in Section 5 and Appendix B, action-dependent normalization can reweight the expectation. This is an important theoretical limitation; however, we did not observe "catastrophic unlearning" in any experiments. Performance typically improved or flattened asymptotically.
>
> We do not claim to have resolved this issue in the current paper. Our claim is narrower: we make the phenomenon explicit, distinguish it from the milder state-reweighting effect in value learning, and show that it does not prevent strong empirical performance in the streaming benchmarks studied here. Appendix B also points to a principled next step: action-independent intentional rules, where the step size may depend on the state but not on the sampled action.
>
> ---
> ### 3. Cross-Sample Effects from Shared Parameters
> We agree this is a real concern with shared nonlinear function approximation. A step chosen to achieve the intended change on the current sample can induce unintended changes at other states where the network is more sensitive. This is exactly the issue discussed in Appendix C.
>
> Appendix C also proposes a simple safeguard: prevent the denominator from becoming much smaller than its typical scale. Empirically, this safeguard did not improve results in the benchmarks studied here, suggesting that cross-sample effects were not a dominant failure mode in these experiments. Still, we agree they may matter more in larger or more nonlinear architectures.
>
> ---
> ### 4. On Clipping the Final Step Size $\alpha_t$
> We agree that directly clipping $\alpha_t$ is a principled safeguard against rare extreme steps caused by unusually small denominators, and we already mention such caps in the limitations discussion (e.g., see line 418, right column).
>
> At the same time, we view $\alpha_t$ clipping as complementary to, not a replacement for, $\delta_t$ clipping. The two address different issues: $\alpha_t$ clipping limits step magnitude, whereas $\delta_t$ clipping limits propagation of unreliable learning signals. In the actor, a large advantage can come from critic error; in the critic, a large TD error can come from an inaccurate bootstrapped target. We also now add a $\delta$-clipping ablation: removing it changed final return only slightly (see our response to the first comment of Reviewer t6dB). This suggests it is not the main source of stability here, but remains a reasonable safeguard in higher-variance settings.
>
> More broadly, while safeguards like  $\alpha_t$-clipping would likely improve performance, we view the "lack of extensive safeguards" as **part of the paper’s positive message.** The present method uses a fairly strong form of normalization, based on the squared gradient norm and with relatively little additional protection, yet it still works surprisingly well in the streaming deep RL setting studied here. We think that is an important result in itself. It suggests that the intentional-update approach is robust enough to be useful even in this relatively aggressive form, and it gives credibility to a broader line of future work exploring more conservative and more refined variants, which may be especially valuable in deeper and more nonlinear networks.

---

> > ### Author Rebuttal · Reviewer_nhSU · 2026-04-01
> >
> > The authors' response was helpful and insightful.
> >
> > I am satisfied with the rebuttal and will maintain my positive score.

---

### Official Review · Reviewer_2mL9 · 2026-03-12

**Soundness:** 3
**Presentation:** 2
**Significance:** 2
**Originality:** 2
**Overall Recommendation:** 4
**Confidence:** 3

**Summary:**

This paper focuses on intentional updates within the streaming reinforcement learning (RL) setting. Typically, the step-size for gradient based methods is selected in some parameter space which can yield inconsistent changes in the model's outputs. The authors propose intentional updates where given the intended outcome of an update, they solve for the step size that essentially gives that outcome hence the name intentional updates. They study intentional updates within the streaming deep RL setting and focus on intentional TD (temporal difference), intentional Q-learning, and intentional policy gradient. Additionally, they conduct a holistic empirical evaluation showcasing the improved stability of their method.

**Compliance With Llm Reviewing Policy:**

Affirmed.

**Final Justification:**

The authors have acknowledged all the points that I brought up in my review, and I am convinced by their explanations. Thus, I am raising my score.

**Key Questions For Authors:**

1. In the policy learning setting, intentional updates induce action-dependent step sizes, which effectively reweight the policy-gradient contributions of different sampled actions. Could the authors explain more carefully why this reweighting does not appear to harm learning in the empirical evaluations?

**Limitations:**

Yes

**Strengths And Weaknesses:**

Strengths:
-The paper tackles an important problem within the streaming RL setting.
-The core premise of the paper is nicely stated and easy to understand: deriving a step-size that achieves some level of desired change in the output space has clear reasoning over picking a value in the raw parameter space for the learning rate.
-The empirical evaluations are quite broad spanning continuous control, discrete control, and prediction.


Weaknesses:
-The technical depth of the main contribution seems a bit limited. The main step-size rule is closely linked to a first-order Taylor argument/normalized LMS so the technical novelty itself is limited. To me, the technical contribution is mainly repackaging step-size control for streaming RL versus introducing a new optimization principle.

---

> ### Author Rebuttal · Authors · 2026-03-31
>
> We thank the reviewer for their valuable feedback, giving us the opportunity to better clarify the paper’s contribution.
>
> ### 1. On Technical Depth, Contributions, and Novelty Beyond NLMS
>
> We agree that the intuition is rooted in normalized methods such as NLMS, as stated in the abstract and introduction. The contribution lies in making the intentional principle work in streaming RL, where the relevant objects, constraints, and failure modes are quite different from supervised learning.
>
>
> * **The main technical contribution is the RL instantiation.** In RL, one must decide both **what** to control and **how** to control it, as there is no predefined loss function like those in supervised learning. For value learning, we target a fractional reduction of the TD error. For policy learning, we target sampled log-probability change as a cheap streaming proxy for local policy change. These are not direct carry-overs from NLMS.
>
> * **Several RL-specific difficulties arise that are absent in standard NLMS.** The trace extension is one example that is genuinely nontrivial. With eligibility traces, one update affects a whole fading history of predictions, not just the current sample. This means the intended outcome must be defined in terms of an *aggregate change*, not a pointwise one. A naive normalization with traces does not recover the desired scaling; the construction in the paper was designed precisely to address this issue. Please see the last paragraph of Section 4 for further discussion and intuition on why this design is not straightforward. A related issue arises with RMSProp-style entrywise scaling: once the update direction is changed by a diagonal preconditioner, the intentional scalar step size must be re-derived for that transformed direction if it is to retain its meaning in function units. These are not details present in standard NLMS, and addressing them is part of the technical contribution.
>
> * **The contribution is also one of bringing an old idea into a new and difficult setting.** NLMS is classical in linear adaptive filtering and control, but it is far from standard practice in modern deep RL and, more broadly, in much of current AI. Part of the contribution here is to show that this style of function-space step-size control can be practical and effective in today’s nonlinear RL setting.
>
> * **We believe the impact is larger than a simple repackaging.** Streaming deep RL is a particularly hard regime, and recent work has emphasized that many successful batch or replay-based methods break down when pushed to the fully streaming setting (Elsayed et al. 2024). In that context, showing that intentional step-size control can work robustly, efficiently, and at scale is, in our view, a meaningful contribution.
>
> * **It also opens several directions for future work.** The present paper uses likelihood-ratio policy gradients, yet already gets surprisingly close to strong replay-based methods in continuous control. Extending the same intentional-update principle to reparameterized actor updates is a particularly promising next step. More broadly, the paper opens the door to a family of function-space step-size methods for modern RL. Several such directions are discussed in Section 9 of the manuscript.
>
> So we agree with the reviewer on one part of the framing: the contribution is not a brand-new optimization principle. But we would push back on the idea that it is merely a repackaging. The technical work lies in identifying the right controlled quantities for RL, designing the corresponding update rules, and making them work in the streaming deep RL regime. We will revise the paper to make this claim sharper and more explicit.

---

> > ### Author Rebuttal · Reviewer_2mL9 · 2026-04-03
> >
> > I thank the authors for their detailed and well-argued rebuttal. I find the clarification on the eligibility trace extension convincing, and addressing aggregate change is genuinely non-trivial. I also appreciate the honest framing that the contribution is not a new optimization principle but rather making an old one work in a difficult and practically important setting. In my opinion, this still constitutes an important contribution. The breadth of the empirical evaluation supports this claim. With the response of the rebuttal, I will increase my score.

---

### Official Review · Reviewer_V7a2 · 2026-03-12

**Soundness:** 3
**Presentation:** 3
**Significance:** 2
**Originality:** 3
**Overall Recommendation:** 4
**Confidence:** 2

**Summary:**

The paper proposes intentional updates, a method for stabilizing streaming reinforcement learning by controlling the magnitude of changes in model outputs rather than directly tuning parameter updates. The approach derives step sizes from desired changes in predictions or policy outputs and applies this idea to value learning and policy gradient methods. Experiments on several standard RL benchmarks suggest that the proposed method improves learning stability in streaming settings while maintaining competitive performance with existing baselines.

**Compliance With Llm Reviewing Policy:**

Affirmed.

**Key Questions For Authors:**

Please refer to the weakness section above.

**Limitations:**

yes

**Strengths And Weaknesses:**

### Strengths
- I quite like the core idea of the paper, which is intuitive and easy to understand. Instead of directly tuning the learning rate, the method controls the magnitude of changes in the model output, providing a clear and appealing perspective on step-size selection.
- The paper addresses an important issue in streaming or online RL settings, where single-sample updates can lead to instability and overshooting. Framing the problem in terms of controlling prediction or policy changes is well motivated.
- The proposed method is relatively simple to implement. It only requires computing gradient norms and applying a normalization factor, which avoids the heavier computations required by methods such as natural gradient or trust-region approaches.
- The empirical evaluation is conducted on several standard RL benchmarks, including continuous control tasks and Atari-style environments. The paper also includes ablation experiments that help clarify the contribution of certain design components.

&nbsp;

### Weaknesses
- While the paper replaces the traditional learning rate with a change-scale parameter (η), which makes the update magnitude more interpretable, this essentially replaces one hyperparameter with another (α → η). In practice, η still needs to be tuned.
- By explicitly constraining the magnitude of policy updates, the method may limit rapid policy changes. This could potentially make exploration more difficult in environments with sparse rewards or long horizons. I am curious about the authors’ perspective on this issue.
- The derivation relies on a first-order Taylor approximation of output changes. Since neural networks are highly nonlinear, the actual change in the model output may deviate from this approximation. The paper also does not provide theoretical convergence guarantees.
- While the paper reports results across multiple tasks, the discussion of the experimental findings is quite limited. Intentional AC generally achieves strong and stable performance, but in some environments (e.g., Finger-turn in Figure 2) it performs the worst among the compared methods. It would be helpful if the authors could provide some insight into why the method works particularly well on tasks such as Dog but performs relatively poorly on tasks like Finger.
- **(Minor)** The descriptions of Figures 7.3, 7.4, and 7.6 are on page 6, whereas the figures themselves appear on pages 7 and 8, which reduces readability. Placing the figures closer to their discussion would improve clarity.

---

> ### Author Rebuttal · Authors · 2026-03-31
>
> We thank the reviewer for their insightful and constructive feedback. TThe comments helped us clarify both the scope of the paper and the empirical evidence supporting the method.
>
> ---
> ### 1. Interpretability and Tuning of $\eta$
> We agree that $\eta$ is still a meta-parameter. The key difference is that it is chosen in **function units**, not parameter units, which makes it easier to interpret and transfer.
>
> * **Transferability:** We did not tune per environment. For IntentionalAC, we used one shared setting across all MuJoCo and DM Control tasks: $\eta_{\text{critic}}=0.5$ and $\eta_{\text{actor}}=0.05$. For Intentional Q-learning, we used a single $\eta=0.25$ across both MinAtar and Atari.
> * **Predictable range:** Unlike a conventional stepsize, whose useful scale can vary by orders of magnitude with architecture or reward scale, $\eta$ has a narrower and more intuitive range because it represents a fractional change in the target quantity.
> * **Architectural Invariance:** A conventional learning rate is tied to parameterization. By contrast, $\eta$ is grounded in the functional units of the agent's behavior. We expect this to provide a significant degree of invariance across network architectures; whether a model is wide or deep, the meaning of a "5% change in action probability" remains constant.
>
> So $\eta$ is not “free,” but it is easier to choose, easier to transfer, and easier to understand than a conventional stepsize. We will sharpen this discussion in the revision.
>
> ---
> ### 2. Exploration and Sparse Rewards
> The method constrains the **average** update scale, **not the maximum**.
> * **Relative Scaling:** For the actor, the intended change is proportional to $A_t / \bar A_t$, where $\bar A_t$ is a running average of advantage magnitude. In sparse-reward settings, most updates have small advantage, so $\bar A_t$ stays small. When a rare informative reward arrives, $|A_t| / \bar A_t$ can become large, allowing a correspondingly large policy update. In that sense, the method can still respond strongly to rare useful experience.
> * **Eligibility traces also help here:** in the streaming setting, they allow a rare reward to propagate backward through several recent decisions in a single update, improving credit assignment without replay. We will add this discussion to the revision.
>
> ---
> ### 3. Validity of the First-Order Approximation
> To address concerns about nonlinearity, we measured the **Update Fidelity Ratio** (defined as Actual Change divided by Intended Change) across 5M steps in MuJoCo tasks. The ratio is tightly concentrated near **1.0**, confirming that first-order Taylor expansion is a  reliable proxy for function change in the networks used in this paper.
>
> *Table 1: Statistics of Update Fidelity Ratio (averaged over 30 independent runs on Ant, Humaniod, and HumaniodStandup)*
> | Network | Std | 1st Percentile | 99th Percentile |
> | :--- | :---: | :---: | :---: |
> | **Policy** | 0.021 | 0.949 | 1.031 |
> | **Value** | 0.039 | 0.898 | 1.051 |
>
> In this test we used $\lambda=0$ rather than traces, because the trace case uses a conservative approximation and would make the fidelity test less interpretable. While deeper networks may require further safeguards (like the $\alpha$-clipping discussed in Section 9), the approximation holds well for standard RL architectures.
>
>
> ---
> ### 4. Task-Specific Performance Differences
> We agree that the task-specific pattern deserves explanation. We ran additional experiments to investigate Finger-turn, where IntentionalAC under the shared setting performs worse than on tasks such as Dog.
>
> The main finding was that the $\eta$ parameters in the paper were too large for Finger-turn, resulting in unstable training across many seeds. Performance improved substantially when we reduced the intentional scales to $\eta_{\text{actor}}=0.01$ and $\eta_{\text{critic}}=0.1$, which are both 5x smaller than the paper. This suggests that the weaker Finger-turn result is not a failure of the method itself, but a case where the shared transferable setting is too aggressive for that environment.
>
> This also matches the goal of Figure 2: we intentionally reused the same setting from MuJoCo across the full DM Control suite to test transferability, not per-task optimality. Finger-turn appears to be an outlier under that shared setting, while Dog lies closer to its sweet spot. We will add this discussion to the revision.
>
> ---
> ### 5. Figure Placement
> We agree. In the revision, we will move Figures 7.3, 7.4, and 7.6 closer to the corresponding discussion to improve readability and flow.

---

> > ### Author Rebuttal · Reviewer_V7a2 · 2026-04-04
> >
> > Thank you for the detailed rebuttal. My concerns have been addressed, and I will maintain my positive rating.

---

### Decision · Program_Chairs · 2026-04-30

**Decision:**

Accept (regular)

**Comment:**

The paper proposes a simple yet effective idea for improving the stability of streaming reinforcement learning: instead of prescribing a fixed stepsize, it determines the update magnitude based on the outcome of the desired update itself. The experiments demonstrate the effectiveness of this approach. The authors were able to address the reviewers’ concerns regarding the significance of the idea, while also providing additional experiments in response to further requests. Overall, this is a solid contribution to the area, and I therefore recommend accepting the paper.